# Configuring Parallel Training of Neural Networks using Bayesian Optimization

## Abstract

Training of modern large neural networks (NNs) is often done in parallel across multiple GPUs. While existing parallel training frameworks easily allow NN training using multi-dimensional parallelism, the challenge remains in finding the optimal hyperparameters, such as the best balance between the sizes of each parallelism dimension, which would result in the highest training throughput. Due to a large number of candidate parallelism configurations (PCs) for a given training scenario, it is infeasible to perform exhaustive search over all of them. Existing PC optimization methods typically either require conducting training trials on a large number of PCs, each of which can be expensive to perform, or rely on an approximate cost model which may be inaccurate and hardware-specific. To overcome these issues, we present OPPA, which combines constrained Bayesian optimization methods with prior knowledge in the form of a parallelism-informed prior belief, to obtain an optimal PC using a minimal number of NN training trials. We also propose a framework for early termination of trials involving suboptimal PCs, whose efficiency gains can be theoretically justified. We show that OPPA finds an optimal PC more efficiently for training transformers on various multi-GPU systems compared to the methods used in existing parallel training frameworks.

## 1 Introduction

Modern advances in deep learning have arisen from the ability to scale neural networks (NNs) to larger sizes. In natural language processing, for example, transformer-based models (Vaswani et al., 2017; Devlin et al., 2019), large language models (LLMs) (Touvron et al., 2023; OpenAI et al., 2024) and multimodal models (Radford et al., 2021; Liu et al., 2023), composed of millions or even billions of parameters, have shown tremendous success in tasks such as text classification, text generation, and language understanding. Due to their size, these large NNs often cannot be trained on standard machines with a single processor. To scale up the training process, it is necessary to distribute the NN training workload across a cluster of machines and parallelize the training process. Different parallelism methods for NN training have been proposed, including data parallelism (Rajbhandari et al., 2020; Zhao et al., 2023), pipeline parallelism (Huang et al., 2019; Narayanan et al., 2019), tensor parallelism (Shoeybi et al., 2020), and combinations of these three parallelism methods also referred to as *multi-dimensional parallelism* (Rasley et al., 2020; Shoeybi et al., 2020; Li et al., 2023).

In NN training, to fully utilize the given hardware and reduce the computation time, we would like to maximize the *throughput* of training, or the number of training steps processed in a given time. The throughput will depend on the selected *parallelism configurations* (PCs), which in large-scale parallel training frameworks (Kuchaiev et al., 2019; Rasley et al., 2020; Shoeybi et al., 2020; Li et al., 2023) are specified by several hyperparameters which would also include the size of each parallelism dimension. In practice, it is *difficult to accurately quantify how the choice of PC affects the training throughput*, as it would depend on the NN architecture, the training data, the hardware, or the exact implementations of the parallel training framework. While there are works on approximating (and optimizing) the training throughput of a PC (Li et al., 2022; Zheng et al., 2022; Zhang et al., 2024), *these approximations require strong assumptions* on the compute hardware and the specific parallelized NN training implementation and may not capture all nuances of a parallel training instance, and so relying on them alone may not be reliable enough to directly inform the optimal PC to select.

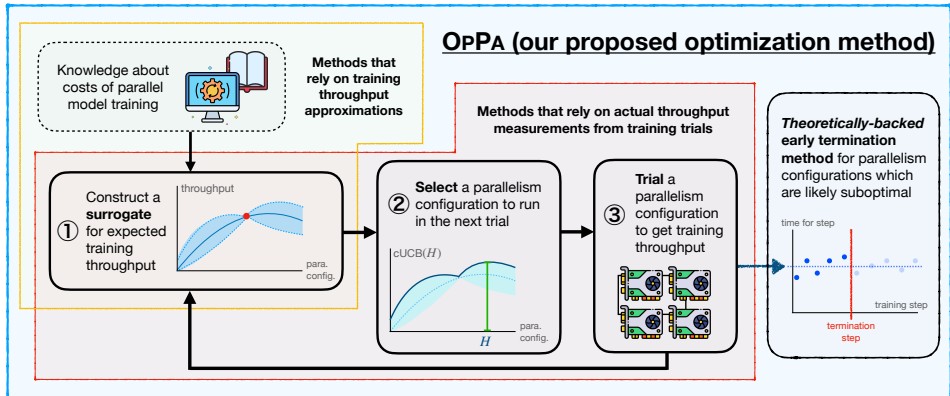

Figure 1: Main idea of OPPA. OPPA combines Bayesian optimization with knowledge on parallel model training to find the parallelization strategy which achieves the highest training throughput.

In practice, *the most reliable method to consider all possible factors during parallel training would be to conduct real training trials* with each PC on real hardware. Unfortunately, due to the large number of possible PCs, *performing an exhaustive search would be extremely inefficient*. To circumvent this, existing parallel training frameworks will use methods to select a subset of candidate PCs to trial. However, these methods are still inefficient due to simplistic optimization algorithms which are *unable to adapt to known training throughput measurements* and *do not utilize on any prior domain knowledge*, therefore still need to run excessively many trials to find a good PC candidate.

To efficiently select the PC that achieves the best throughput, we therefore need the ability to adaptively select potentially good PCs to trial, while also filtering out poor candidates using information from trialed PCs and from existing domain knowledge. Given these considerations, it may be possible to use black-box optimization methods such as Bayesian optimization (BO) (Gelbart et al., 2014; Frazier, 2018). However, *naive application of BO is still inefficient*, and could be improved if characteristics of PC trialing and the parallel training process are taken advantage of by the algorithm.

In this paper, we introduce the O̲P̲TIMIZER FOR P̲A̲RALLELISM CONFIGURATIONS, abbreviated as OPPA, which *adaptively and rapidly optimizes the PC for efficient parallel training*. To achieve this, OPPA boosts the efficiency of BO by novelly exploiting *early termination of trials involving suboptimal PCs* and a *parallelism-informed prior belief*. Despite incorporating these efficiency tricks, OPPA still theoretically guarantees sublinear regret. The main idea of OPPA is presented in Fig. 1. In Sec. 3, we first formulate the problem of finding the optimal PC as a black-box function optimization problem with black-box constraints. In Sec. 4, we discuss the design choices of OPPA. We develop a surrogate model with a *parallelism-informed prior belief* based on knowledge from parallelized NN training that can generalize to many hardware setup and training scenarios (Sec. 4.1), which is then used to select promising PCs to trial using constrained BO (Sec. 4.2). We also discuss the process of trialing a PC and *propose a novel BO technique which early terminates trials with suboptimal PC*, with both theoretical and empirical justification (Sec. 4.3). Finally, we empirically demonstrate the effectiveness of OPPA in Sec. 5, showing that OPPA can more efficiently find a good PC for training transformers compared to existing methods and compared to naively using BO without modifications.

## 2 BACKGROUND AND RELATED WORKS

In this section, we provide an overview of current techniques of parallelized model training on multiple GPUs, and how optimal parallelism configurations are currently found. We also provide a brief overview of Bayesian optimization, which is a technique we will use in our proposed method.

### 2.1 PARALLELIZED NEURAL NETWORK TRAINING

To effectively train large neural networks (NNs), the training workload can be distributed across multiple GPUs. Different parallelism dimensions split the workload differently, which affect the

amount of computation per GPU, amount of communication between each GPU, and the amount of memory required in each GPU. Here, we briefly discuss some of these existing parallelism techniques.

**Data, tensor, and pipeline parallelism.** The most basic method to parallelize NN training is data parallelism (DP) (Li et al., 2020), where a batch of training data is split and distributed to each device, separately processed by the local model replica, before gathering the gradients from each device. While DP is simple, it replicates the model on each device, taking up additional memory. To solve this, techniques such as the Zero Redundancy Optimizer (ZERO) (Rajbhandari et al., 2020) in the DEEPSPEED package, or Fully-Sharded Data Parallel (FSDP) (Zhao et al., 2023) have been proposed to perform some sharding of model parameters or gradients to avoid full model replication. Furthermore, tensor parallelism (TP) (Shoeybi et al., 2020; Bian et al., 2021) and pipeline parallelism (PP) (Huang et al., 2019; Narayanan et al., 2019) have been proposed which partition, respectively, the tensors in the model and the model execution pipelines onto multiple devices. The specific implementations of DP, TP, PP can often also be further controlled by the user, including factors such as which tensors are sharded or how many shards they are partitioned into, which can affect the overall throughput. The three types of parallelism are discussed further in App. A.

**Multi-dimensional parallelism.** Many frameworks (Rasley et al., 2020; Shoeybi et al., 2020; Li et al., 2023) have also been developed to combine the use of DP, TP, and PP within the same training process. These frameworks provide simple interfaces for users to specify the desired *parallelism configuration* (PC), which includes hyperparameters controlling the execution of the parallel training process, a subset of which specifies the size of each parallelism dimension. These frameworks then automatically handle tensor sharding and execute the parallel training pipeline as per the specified PC. These frameworks may also manage training on multi-node setting or even heterogeneous hardware. While these frameworks allow practitioners to easily specify a PC for training, *selecting the optimal PC for the most efficient training is difficult*, since the optimal PC will non-trivially depend on the GPU specifications, communication bandwidth of the GPU devices, the specific NN architecture or the training data (Li et al., 2023; Lin et al., 2024; Wagenländer et al., 2024). For example, DP is ineffective for large models or large batch sizes, since the additional model replications may cause out-of-memory errors. Meanwhile, PP is less effective on smaller models, as communication costs between each pipeline stage may dominate the computation of the fragmented pipeline.

**Optimization of multi-dimensional parallelism configuration.** The most accurate way to find the optimal PC would be to trial all possible PCs on the actual training hardware to determine which one results in the highest training throughput. However, this is prohibitively expensive since there can be a large number of possible PCs, and each trial would itself require computational resource and time which may be limited on real clusters. To circumvent this, frameworks such as NEMO[1] (Kuchaiev et al., 2019) and DEEPSPEED[2] (Rasley et al., 2020) have implemented methods for automatic PC tuning based on running NN training trials for a few training steps on a number of PCs. The PCs trialed are often either selected non-adaptively (e.g., based on random selection), or adaptively based on a simple surrogate function. However, these methods are unable to *efficiently use the measured throughput of trialed PCs* to model the true throughput and perform informed optimization, and therefore still require a large number of training trials to obtain a good PC.

Since running training trials may be expensive, we may consider constructing a surrogate model to approximate the computation and communication costs for different parallelism strategies (Li et al., 2022; Zheng et al., 2022; Zhang et al., 2024), which would allow us to use domain knowledge to filter out suboptimal PCs while performing fewer trials, or even by not trialing any PCs at all. This methods, however, would *require an implicit assumption that the surrogate of the true training throughput is correct*, which may not always be possible because surrogates may be unable to fully capture the nuances of practical parallel training implementations. Furthermore, a fixed surrogate model would not be easily extendable to new hyperparameters or parallelism nuances which may arise in a PC, which is important especially with the ever-growing parallelism training literature.

## 2.2 BAYESIAN OPTIMIZATION

In order to more efficiently select a PC to trial and to optimize for, we will utilize Bayesian optimization (BO) (Frazier, 2018). BO aims to maximize some black-box function $f : \mathcal{X} \to \mathbb{R}$ which is often

---

[1]https://docs.nvidia.com/nemo-framework/user-guide/latest/usingautoconfigurator.html
[2]https://www.deepspeed.ai/tutorials/autotuning/

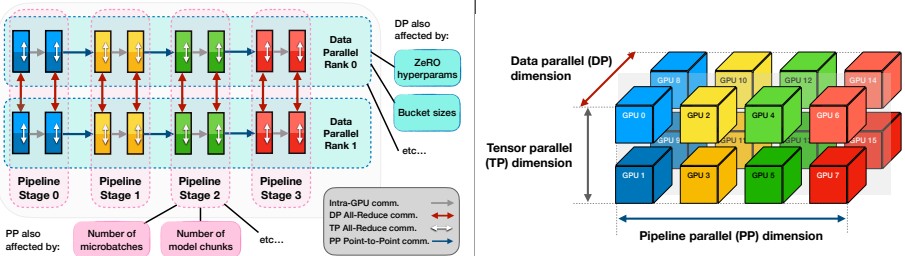

Figure 2: *Left:* Visualization of a parallelism configuration with the hyperparameters that can be tuned. *Right:* Visualization of GPU allocation for 3D parallelism according to the dimension sizes.

expensive to query and whose derivative is unknown. The black-box function is modeled a Gaussian process (GP) (Rasmussen & Williams, 2006), which is characterized by a prior mean $\mu_{\text{prior}}(\cdot)$ and a kernel function $k(\cdot, \cdot)$. Given a set of observations, we perform Bayesian inference to obtain a posterior GP, which is made up of a posterior mean and posterior covariance, encoding the expected value and the uncertainty of the function respectively. With the posterior GP, the BO procedure selects an input that maximizes some acquisition function, such as the expected improvement (Jones et al., 1998) and the upper confidence bound (Srinivas et al., 2012). These acquisition functions balance between exploring unique inputs that have not been queried to obtain some function estimate, and exploiting inputs likely to have high function values in order to efficiently recover a global optimum of $f$. We provide a more technical overview of GP modeling and BO in App. B.

BO is a widely used to optimize black-box functions which have no closed form and are expensive to evaluate. This include a wide range of problems, such as experimental design (Lei et al., 2021; Rainforth et al., 2023) or material design (Zhang et al., 2020). More relevant to our work, BO is also commonly used for optimizing the NN architecture such that achieves the best performance in a given task (Snoek et al., 2012). Finding the optimal PC, however, differs in two aspects. First, the hyperparameters in a PC on the training throughput have better-defined mechanics (even if not completely known), which can be partially described based on domain knowledge. Modeling via a GP allows incorporation of these knowledge through a good choice of prior belief, which reduce the number of trials required. Second, trialing a PC allows for repeated measurements from many sequential training steps. Here, running many training steps is costly and may be redundant, and so adaptive early termination may be applicable to reduce the number of training steps needed to be run.

## 3 PROBLEM SETUP

In this section, we describe the problem setup. For our problem setting, we consider a parallelism configuration (PC), visualized in Fig. 2, which contains a list of tunable hyperparameters found in typical parallel training frameworks, and controls various aspects of parallelized NN training. As shown, a subset of these hyperparameters determines the *size of each parallelism dimension*. In our paper, we consider 3D parallelism where we use dp, tp, and pp, to indicate the size of the data, tensor, and pipeline parallelism dimensions respectively. We assume that their product $\text{dp} \cdot \text{tp} \cdot \text{pp}$ is equal to the number of available GPUs n_gpus. The remaining hyperparameters determine the *specific implementations of each parallelism dimension*, which may include hyperparameters of the ZERO optimizer which controls the DP implementation, the number of microbatches and model chunks which control how the PP implementation, or other hyperparameters specific to the parallel training framework. We discuss these hyperparameters further in App. C.1.

We let $\mathcal{H}$ be the set of all possible PCs. The goal of our problem is to find the optimal PC $H^* \in \mathcal{H}$ which results in the *highest throughput* (i.e., can run the most number of training steps per unit of time), while *fitting in the GPU memory* (i.e., the maximum GPU memory required is less than $M_0$). For some PC $H$, we let $\mathcal{R}(H)$ and $\mathcal{M}(H)$ be, respectively, the throughput and the maximum memory usage when using PC $H$. Then, our problem of finding the optimal $H^* \in \mathcal{H}$ can be formulated as a constrained maximization problem given by

$$\underset{H \in \mathcal{H}}{\text{maximize}} \ \mathcal{R}(H) \quad \text{s.t.} \quad \mathcal{M}(H) \leq M_0. \tag{1}$$

To evaluate a PC $H$, we can perform a short training trial to estimate its throughput and maximum memory usage. To estimate the throughput of $H$, we can measure the times $t_1, t_2, \ldots, t_q$ over $q$ training steps, which can then be used to approximate the throughput as $\mathcal{R}(H) \approx q^{-1} \sum_{j=1}^{q} t_j^{-1}$. As we demonstrate in Fig. 3, the time to execute each training step can vary, for example due to setup and compilation processes at the start of the training, and unpredictable system fluctuations which occur throughout training. Therefore, it is unreliable to predict the training throughput using only few training steps, but rather should aggregate the values across multiple training steps. While we can choose to perform $q_{\max}$ training steps during the trial, we can also choose to terminate the trial early after fewer than $q_{\max}$ training steps, although the estimate of $\mathcal{R}(H)$ may also be more inaccurate. To measure the maximum allocated GPU memory throughout the training steps, using CUDA-based PYTORCH, this can be done using the `torch.cuda.max_memory_allocated()`

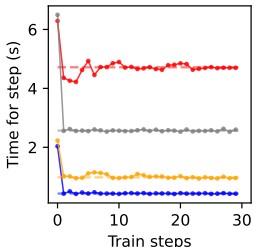

Figure 3: Time for each training step for a few training trials with different PCs. Each color represents a different PC used, and the dashed line represents the throughput estimate for that PC.

function, which records the maximum allocated GPU memory achieved at any point during training.

When designing our algorithm, we also note two additional characteristics of the problem:

- $\boxed{\text{A}}$ Since $\mathcal{R}$ and $\mathcal{M}$ are dependent on many factors which may be difficult to model or even known exactly, we assume that *their exact forms are only partially known* given the existing domain knowledge. Therefore, a good surrogate for $\mathcal{R}$ and $\mathcal{M}$ should be able to incorporate domain knowledge with some uncertainty modeling based on observed training trials.

- $\boxed{\text{B}}$ Even though a PC can be (and should be) trialed on real hardware, *running a single trial incurs a high cost*. This is especially true with suboptimal PCs since the same number of training steps on a suboptimal PC would require more time to execute. This motivates us to design an optimization algorithm such that only promising PCs are trialed, and PCs which are likely suboptimal are not trialed or are only trialed for a shorter period of time.

## 4 METHOD

In this section, we describe OPPA, which incorporates BO with domain knowledge on training parallelism to perform an informed selection of the optimal PC for parallel training. As shown in Fig. 1, OPPA alternates between three steps; ① modeling the training throughput and maximum memory usage based on observed data using a GP with a parallelism-informed prior belief, ② finding the best PC to trial next using BO, and ③ conducting NN training trials for some number of training steps to obtain an estimate of the training throughput and maximum memory usage for a selected PC.

### 4.1 CONSTRUCTING A SURROGATE MODEL

In Step ①, we attempt to construct a surrogate model to predict the throughput and the memory usage. As suggested in $\boxed{\text{A}}$, to explicitly model the imperfections in our domain knowledge, we assume that the true throughput $\mathcal{R}$ and maximum memory usage $\mathcal{M}$ can be decomposed as

$$\mathcal{R}(H) = \hat{\mathcal{R}}(H; \theta_{\mathcal{R}}) + f_{\mathcal{R}}(H), \qquad \mathcal{M}(H) = \hat{\mathcal{M}}(H; \theta_{\mathcal{M}}) + f_{\mathcal{M}}(H), \qquad (2)$$

where $\hat{\mathcal{R}}$ and $\hat{\mathcal{M}}$ represent the parallelism-informed prior beliefs with hyperparameters $\theta_{\mathcal{R}}$ and $\theta_{\mathcal{M}}$ constructed based on the domain knowledge on parallel training, and $f_{\mathcal{R}}(H)$ and $f_{\mathcal{M}}(H)$ additional unknown contributions not captured by our domain knowledge.

**Parallelism-informed prior belief.** The functions $\hat{\mathcal{R}}$ and $\hat{\mathcal{M}}$ aim to estimate $\mathcal{R}$ and $\mathcal{M}$ respectively based on existing knowledge about parallel NN training. We *do not require $\hat{\mathcal{R}}$ and $\hat{\mathcal{M}}$ to be completely accurate*, but instead only be reasonable estimates and generalize across multiple training scenarios.

For OPPA, to approximate the throughput, we consider the time per training step for the computation $\hat{\mathcal{T}}_{\text{comp}}$ and for the communication $\hat{\mathcal{T}}_{\text{comm}}$, which can combine to approximate the throughput as $\hat{\mathcal{R}}(H; \theta_{\mathcal{R}}) = \left[\hat{\mathcal{T}}_{\text{comp}}(H; t_{\text{c}}) + \hat{\mathcal{T}}_{\text{comm}}(H; \mathbf{C})\right]^{-1}$ where $\theta_{\mathcal{R}} = \{t_{\text{comp}}, \mathbf{C}\}$ are learned hyperparameters.

For $\hat{\mathcal{T}}_{\text{comp}}$ we consider the additional computation time that arise from the pipeline bubble in PP Narayanan et al. (2019). Meanwhile, inspired by Xiong et al. (2024), for $\hat{\mathcal{T}}_{\text{comm}}$, we consider an idealized training scenario visualized in Fig. 2 which assumes the model to compose of roughly identical blocks, and considers the communication from the All-Reduce operations involved in DP and TP, and point-to-point communications involved in PP. We consider a canonical ring/tree scaling with hierarchical aggregation such that intra- and inter-host connections are separately modeled, with the constants for the specific communication types collapsed into $\mathbf{C}$. Further details of this are given in App. D.1. To model maximum memory usage $\hat{\mathcal{M}}$, we consider the memory required per GPU to store the NN parameters and to store gradient values for backpropagation, as we detail in App. D.2.

In both prior belief functions, a key design choice is to make the prior belief function general enough to capture a variety of training scenarios, with *a simple analytical form based on looser assumptions* about the model, hardware, and network, and *using learnable parameters* (as opposed to fixed constants) to capture approximate cost multipliers which depend on the model and training data sizes, and intra- and inter-host network communications, each of which vary across different training scenarios. These design choices allow OPPA *to be applicable to training scenarios involving a wide variety of model and hardware setups*, due to the balance between capturing the general effects of each parallelism dimension, but not being too specific to overfit to any particular training scenario.

**Additional contribution terms.** Due to the incomplete domain knowledge to fully describe a parallel training process, we aim to learn the unaccounted factors $f_{\mathcal{R}}$ and $f_{\mathcal{M}}$ using real training trials. To do so, we model $f_{\mathcal{R}}$ and $f_{\mathcal{M}}$ using Gaussian processes (GPs). The benefit of using a GP is twofold. First, a GP is typically flexible enough to model unknown functions that may not have an analytical form. Second, a GP can quantify its uncertainty, which allows the surrogate to determine how much it knows about the throughput of a certain PC. This allows OPPA to potentially trial PCs whose throughput it is more uncertain about given the trials conducted.

To model $f_{\mathcal{R}}$ and $f_{\mathcal{M}}$, we use a GP with zero mean. This is done since we assume $\mathcal{R}$ and $\mathcal{M}$ should already be "centered" around the prior belief functions, and so the additional contribution terms would only need to model the deviations from the prior belief which will likely be "centered" around zero. For the kernel function $k$, we first embed the PC $H$ via an embedding $e : \mathcal{H} \rightarrow [0, 1]^p$ which maps each PC to a $p$-dimensional vector. Here, we let $e(H)$ be a concatenation of each hyperparameter value in $H$, where each dimension is scaled to be between 0 and 1 according to the feasible values. Given the embedding, we then use the Matern kernel (Rasmussen & Williams, 2006) with $\nu = 5/2$ where the distance between two PCs is the Euclidean distance of their corresponding embeddings, with some kernel hyperparameters $\theta_k$. The equation for the Matern kernel is given in App. D.3.

Given our prior belief and the modeled additional contribution terms, the decomposition in Eq. (2) encodes our belief that $\mathcal{R}$ and $\mathcal{M}$ is drawn from a GP which is given by

$$\mathcal{R} \sim \mathcal{GP}\big(\hat{\mathcal{R}}(\cdot \; ; \; \theta_{\mathcal{R}}), k(\cdot, \cdot \; ; \; \theta_k)\big), \quad \text{and} \quad \mathcal{M} \sim \mathcal{GP}\big(\hat{\mathcal{M}}(\cdot \; ; \; \theta_{\mathcal{M}}), k(\cdot, \cdot \; ; \; \theta_k)\big). \tag{3}$$

Using measurements from the previous $i-1$ trials, we find the optimal hyperparameters $\{\theta_{\mathcal{R}}, \theta_{\mathcal{M}}, \theta_k\}$ by maximizing the marginal log-likelihood (Rasmussen & Williams, 2006), then perform GP regression to obtain the posterior belief for the throughput $\mathcal{N}\big(\mu_{\mathcal{R}, i-1}(H), \sigma^2_{\mathcal{R}, i-1}(H)\big)$ and the memory usage $\mathcal{N}\big(\mu_{\mathcal{M}, i-1}(H), \sigma^2_{\mathcal{M}, i-1}(H)\big)$ for any PC $H$ that has not been trialed, which would be a normal distribution with their respective mean and variance whose form we state in App. D.4.

## 4.2 SELECTING THE NEXT PC TO TRIAL

In Step ②, the next PC to trial is chosen based on the surrogate constructed in ① using BO. The next PC $H_i \in \mathcal{H}$ to trial in round $i$ is chosen to be the PC which maximizes the constrained upper confidence bound (cUCB) (Srinivas et al., 2012; Wilson et al., 2017), which is given by

$$\text{cUCB}_i(H) \triangleq \mathbb{E}_{\hat{r}_{H, i-1}, \hat{m}_{H, i-1}}\big[X_{H, i-1} + \beta_i \big| X_{H, i-1} - \mathbb{E}[X_{H, i-1}]\big|\big] \tag{4}$$

where $\hat{r}_{H, i-1} \sim \mathcal{N}\big(\mu_{\mathcal{R}, i-1}(H), \sigma^2_{\mathcal{R}, i-1}(H)\big)$ and $\hat{m}_{H, i-1} \sim \mathcal{N}\big(\mu_{\mathcal{M}, i-1}(H), \sigma^2_{\mathcal{M}, i-1}(H)\big)$ are sampled from their respective GPs as modeled from ①, and $X_{H, i-1} = \hat{r}_{H, i-1} \cdot \big(1 - \texttt{sigmoid}(\hat{m}_{H, i-1})\big)$. Note that in the case that memory constraint is not violated (i.e., when $\texttt{sigmoid}(\hat{m}_{H, i-1}) \approx 0$), the objective in Eq. (4) can be reduced to the analytical UCB objective (i.e., $\text{cUCB}_i(H) \approx \mu_{\mathcal{R}, i-1}(H) + \beta_i \sigma_{\mathcal{R}, i-1}(H)$). The cUCB criterion considers a balance between *exploration* of

PCs which have not been trialed, and *exploitation* of PCs which are similar to those with already high throughputs (Jones et al., 1998; Gelbart et al., 2014). With this balance, the BO iteration is able to try enough PCs to construct a reasonable surrogate for the functions, while utilizing the remaining computational resources to trial good PC candidates for to achieve optimal training throughput. This allows the optimization to be more guided and more efficient, satisfying the requirement in $\boxed{B}$.

### 4.3 TRIALING THE NEXT PC

Finally, in step ③, we perform a training trial on the PC $H_i$ chosen in ② for $q_{max}$ training steps, and measure the time taken for each training step as $t_{i,1}, \ldots, t_{i,q_{max}}$. We then use these measurements to obtain an estimate of the throughput $\bar{r}_{i,q_{max}}$, where $\bar{r}_{i,q} = q^{-1} \sum_{j=1}^{q} t_{i,j}^{-1}$, with variance $\sigma_{\bar{r}_{i,q_{max}}}^2$ whose computation we detail in App. F.1. In practice, as demonstrated earlier in Fig. 3, the actual measured time for a sequence of training steps may contain outliers which can skew the throughput estimates. As we discuss in App. F.2, to make our estimate more accurate, we remove outlier values of $t_{i,j}$ before computing the throughput estimate. Meanwhile, the maximum memory usage $m_i$ across all training steps is measured by `torch.cuda.max_memory_allocated()`.

**Early trial termination.** In practice, some PCs do not need to be trialed for the full $q_{max}$ steps, since fewer training steps are sufficient to determine that the PC is suboptimal. To save time on these trials, we consider early trial termination, where the training trial only continues if the throughput estimation is above some threshold. More formally, we define an indicator variable $I_{i,q}$ given by

$$I_{i,q} = \mathbb{1}\left[(q \leq q_{min}) \vee \left(\bar{r}_{i,q} \geq \max_{l \in \{1,\ldots,i-1\}} \bar{r}_{l,\hat{q}_l} + \tau_q\right)\right]. \tag{5}$$

Intuitively, $I_{i,q} = 1$ when fewer than $q_{min}$ trials have been conducted, or when the throughput of $H_i$ is likely to be higher than the throughput values found so far. We can continue the $q$th training step as long as $I_{i,q-1} = 1$, and terminate the training trial at the first step $\hat{q}_i$ when $I_{i,\hat{q}_i} = 0$, which is when we are confident that $H_i$ will not improve the best PC we have found so far. Early termination of trials will make suboptimal trials terminate earlier, saving computation time in practice, while still allowing the BO procedure to recover the optimal PC, as we show in the following theorem.

**Theorem 4.1** (Informally stated in terms of $\mathcal{R}$). *There exists some $\{\beta_i\}_{i=1}^N$ and $\{\tau_q\}_{q=1}^{q_{max}}$ such that, with high probability, the cumulative regret is $\sum_{i=1}^{N} \left(\mathcal{R}(H^*) - \mathcal{R}(H_i)\right) = \tilde{\mathcal{O}}\left(\sqrt{N/q_{min}}\right)$, and for all $i = 2, \ldots, N$, if $\mathcal{R}(H_i) < \max_{j \in \{1,\ldots,i-1\}} \mathcal{R}(H_j)$, then $\hat{q}_i < q_{max}$.*

In App. F.3, we prove a more general version of Thm. 4.1, and provide both an intuitive explanation along with empirical justification for early termination. Thm. 4.1 shows that sublinear regret can be achieved even with early termination, which means that OPPA will be able to recover the PC with the best throughput while allowing efficiency gains in practice. Furthermore, it also shows that PCs whose throughput is smaller than those of PCs already trialed will likely have their trials terminated early, therefore allowing OPPA to save resources from trialing suboptimal PCs as mentioned in $\boxed{B}$. We present the pseudocode for OPPA incorporating steps ①, ②, and ③ in App. G.

## 5 EXPERIMENTS

In this section, we present the results for OPPA when used to find the optimal PC for training transformer models on multi-GPU systems. We consider optimizing PC on different transformer-based training scenarios and on different hardware configurations with varying number of GPUs. We focus on transformers since many newer parallel training frameworks are mainly designed for these architectures. Detailed setups for the training scenarios are found in App. H.1.

We compare OPPA with several benchmarks, including RANDOM (random selection), XGBOOST (adaptive selection based on XGBOOST surrogate model (Chen & Guestrin, 2016) and is the current method used by DEEPSPEED (Rasley et al., 2020)), COST-MODEL (method which solely relies on the cost model of the throughput), and VANILLA-BO (which uses BO without any additional modifications). We provide more detailed description of these benchmarks in App. H.2.

We plot the best obtained throughput (in training steps per second) versus how long the optimization has been run, rather than versus the number of PCs that have been trialed, since each trial take a different amount of time to run. Nonetheless, plots for the achieved throughput versus the number of trials run are in App. I.1. The model loss are independent of the chosen PC and thus are not reported.

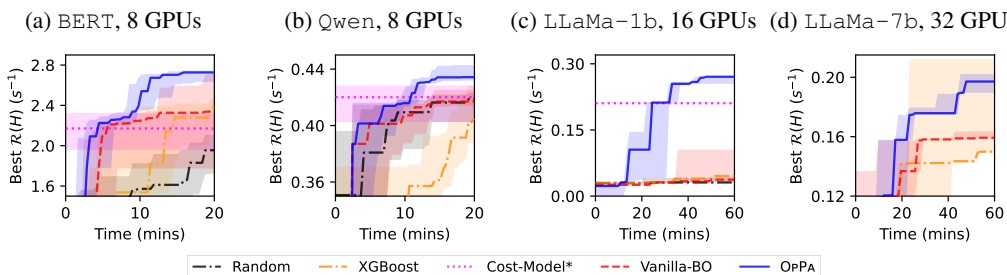

Figure 4: The best obtained throughput (higher is better) versus the duration each algorithm has been run for. The lines show the median across repeated trials, while the error band show the quartiles. Figs. 4a and 4b are both ran on a single host, while Figs. 4c and 4d was ran on multiple hosts. Note that some benchmarks were omitted from Fig. 4d due to computational budget.

**Training on single-host setups.** We first consider training transformers on a single machine with 8 GPUs. In Fig. 4a, we present the results for finding the optimal PC for training the BERT model (Devlin et al., 2019). We see that the methods which use BO outperform non-adaptive and even the other adaptive selection benchmarks. Furthermore, OPPA, which applies a parallelism-informed prior and early termination to BO, is able to achieve better performances than BO alone. We find that OPPA automatically prioritizes PCs with only DP and no ZERO optimizer, which matches our intuition that DP should be adequate for smaller NNs. In Fig. 4b, we consider the Qwen model (Yang et al., 2024), where OPPA again finds a better PC compared to the other benchmarks. Due to the larger model, OPPA now prefers a mix of DP and PP with fewer microbatches to reduce the memory use and synchronization between GPUs. We show the PCs selected by OPPA in App. I.2. We also show the ability of OPPA to optimize the PC on vision models and mixture-of-experts in App. I.3.

To further visualize the efficiency gains of OPPA, in Fig. 5a, we see that OPPA is able to trial many more PCs in a short amount of time compared to other methods due to terminating suboptimal trials early to avoid wasting time. When combined with a parallelism-informed prior belief to efficiently filter out suboptimal PCs, OPPA is able to return a PC with higher throughput, which in turn allows many more training steps to be processed in the subsequent training even after just 20 minutes of PC optimization, as shown in Fig. 5b. This demonstrates the necessity OPPA to achieve faster parallel NN training.

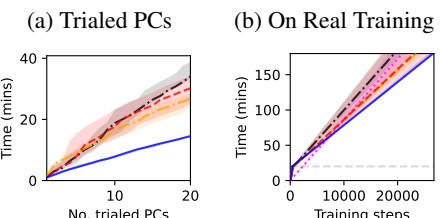

Figure 5: Efficiency of OPPA. Fig. 5a shows the number of PCs trialed by each method. Fig. 5b shows the number of training steps ran by each algorithm during optimization and during subsequent NN training (regions below and above dotted gray line respectively). The legend is the same as in Fig. 4.

**Accuracy of surrogate model.** In Figs. 6a and 6b we compare the throughput predicted by OPPA with the actual throughput scores. We see that even after a few trials, the predictions made by our surrogate already correlate well with the actual throughput. As we progress, the prediction also becomes more accurate, especially among PCs with high throughput where more trials are being run, allowing the optimal PC to be efficiently found. On the other hand, a cost model alone can capture rough trends of $\mathcal{R}(H)$ but not all nuances especially between the better PCs as shown in Fig. 6c, while a GP alone does not allow the surrogate to learn meaningful interpolations of $\mathcal{R}(H)$ as shown in Fig. 6d. In either of these cases, we see that there is a mismatch between the predicted optimal PC and the actual optimal PC. We further demonstrate the quality of the throughput and maximum memory surrogates in App. I.4, and discuss their robustness in App. I.5.

**Effects of each component in OPPA.** In Fig. 7, we performed ablation studies to isolate the effects from each proposed component in OPPA. We see that without early termination, BO would spend more time on suboptimal trials, resulting in a slower search process. Similarly, without the prior belief, we would be less informed about PCs which may be optimal, requiring more time to find the optimal PC. Additional results are presented in App. I.6. We also show the effects of $q_{\min}$ on the performance of OPPA in App. I.7, demonstrating minimal degradation for small $q_{\min}$.

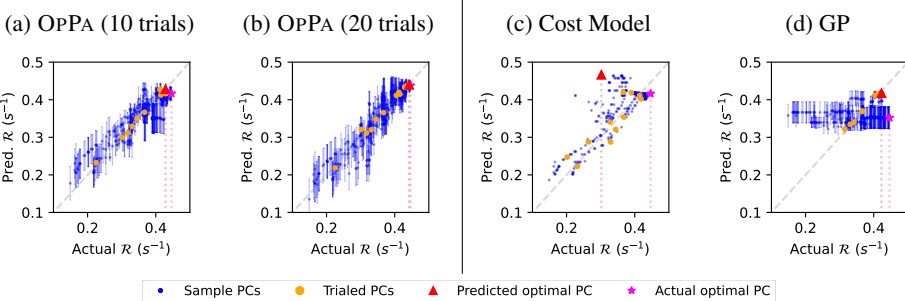

Figure 6: Predicted throughputs from different surrogates versus the measured throughputs for the `Qwen` example. Figs. 6a and 6b represent the surrogate from OPPA after 10 and 20 trials respectively, with error bars showing the uncertainty. Fig. 6c shows the predictions from the cost model alone, while Fig. 6d shows the predictions from using GPs alone (both after 20 trials).

**Training on multi-host setups.** In addition to training on single-host setups, we also tested OPPA on optimizing the PC on multi-host setups. In these cases, the additional communication costs makes the throughput computation less straightforward, while larger number of feasible PCs complicates the optimization problem. We consider two multi-host setups which have different communication types and performance levels. In Fig. 4c, we show the results for tuning the PC for training a LLaMa-3 model (Grattafiori et al., 2024) with 1 billion parameters on a commodity cluster with 16 GPUs, which are setups more commonly found by practitioners with existing hardware in practice. Meanwhile, Fig. 4d are the results for tuning the PC to train a LLaMa-2 model (Touvron et al., 2023) with 7 billion parameters on 32 GPUs distributed across 8 machines in a high-performance computing (HPC) cluster. In both cases, we find that OPPA is still able to outperform other methods by a significant margin. Furthermore, OPPA obtains good PCs consistently (i.e., smaller variance in the resulting throughput), demonstrating its robustness. We also see that BO-based selection methods outperform the other algorithms due to its ability to balance exploration and exploitation, however OPPA is able to do so more efficiently due to the additional guidance from prior knowledge and early termination of trials to save time.

In Fig. 8, we also compare OPPA with cost model-based algorithms, namely AMP (Li et al., 2022) and NNSCALER (Lin et al., 2024). We see that OPPA allow better PCs to be selected compared to methods based solely on optimizing a cost model.

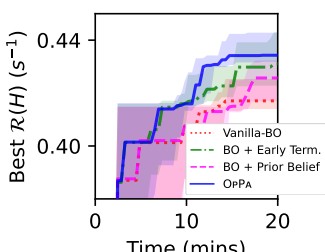

Figure 7: Effect of different components of OPPA on the obtained throughput for the `Qwen` training.

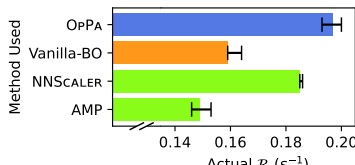

Figure 8: Obtained $\mathcal{R}$ from OPPA compared to those from cost model-based methods for `LLaMa-7b` training with 32 GPUs. Note the results reported for BO and OPPA are the same from Fig. 4d.

This demonstrates the advantages of adaptive methods which effectively incorporate the scores obtained from actual training trials as opposed to solely using cost model surrogates, and the non-trivial modifications made from OPPA which allow for this boost in performance.

# 6 CONCLUSION

We have presented OPPA, which uses constrained Bayesian optimization techniques with a parallelism-informed prior distribution to efficiently optimize the parallelization strategy which can achieve the best training throughput across a variety of hardware configurations. OPPA can be easily applied to other parallel training frameworks due to the minimal assumptions on the implementations of the training parallelism and the simplicity to extend to other hyperparameters. We believe that the parallelism-informed prior belief could be embedded with more prior knowledge on specific training implementation or training of specific NN architectures, which should boost OPPA even further to find a better and possibly more elaborate PC for parallel NN training.

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

## A  ADDITIONAL DISCUSSION ON PARALLELISM TYPES

**Data parallelism.** The most basic type of training parallelism is data parallelism (DP) (Li et al., 2020) where a batch of training data is split into shards and distributed among each device. These shards are then fed into the local replicas of the models, before the parameter updates from each device are synchronized. While simple and often the fastest, the naive DP approach requires replication of the model on each device, which takes up additional storage on each machine. Several methods have since been proposed to perform DP with sharded models, including the Zero Redundancy Optimizer (ZERO) introduced by Rajbhandari et al. (2020) in the DEEPSPEED package, and Fully-Sharded Data Parallel (FSDP) introduced by Zhao et al. (2023). While these frameworks allow for efficient DP implementations, their effectiveness can still heavily depend on the choice of hyperparameters. For example, ZERO involves three different stages of optimization which chooses whether the optimizer states, the model gradients or the model parameters are sharded between each GPU. The choice of sharded items affect the amount of data that has to be stored in each GPU and communicated across GPUs, which in turn affects the throughput of the training and the memory usage in each GPU.

**Tensor parallelism.** Another method for scaling operation is tensor parallelism (TP) where individual tensors are sharded across multiple devices, so that the matrix multiplication operations are instead done in a distributed manner, allowing these operations to scale to larger sizes than otherwise that would fit on a single GPU. TP initially involved splitting a tensor along a single dimension (Shoeybi et al., 2020), however has since also incorporated sharding tensors across multiple dimensions as well (Bian et al., 2021).

**Pipeline parallelism.** In pipeline parallelism (PP) (Huang et al., 2019; Narayanan et al., 2019) we instead partition the model along its execution pipeline, with each model partition running synchronously with microbatches of data. The gradients are accumulated for each microbatch and updated at the end of each training step. By sharding the model and training data into smaller chunks, the GPU memory required at any one time becomes lower, allowing for the training of larger models at the cost of more sequential operation rounds and higher cost of communication between each GPU. The tradeoff between training speed and maximum memory usage can be further controlled based on the size of microbatches and the number of model chunks.

## B  TECHNICAL PRIMER ON GAUSSIAN PROCESSES AND BAYESIAN OPTIMIZATION

In this section, we provide a technical overview of Gaussian process (GP) regression and on Bayesian optimization (BO). The contents are adapted from Rasmussen & Williams (2006); Frazier (2018).

A Gaussian process (GP) $\mathcal{GP}(\mu_{\text{prior}}, k)$ with prior mean $\mu_{\text{prior}}$ and kernel $k$ is a random process where for any subset of input $\mathbf{X}$, its corresponding output is given by a normal distribution $f(\mathbf{X}) \sim \mathcal{N}\big(\mu_{\text{prior}}(\mathbf{X}), k(\mathbf{X}, \mathbf{X})\big)$. The prior mean $\mu_{\text{prior}}(x)$ describes the expected value of the random function $f(x)$ at a certain input, while the kernel function $k(x, x')$ roughly captures the covariance between $f(x)$ and $f(x')$.

Assume we have an unknown function $f$ drawn from the GP. Given a set of observations $D = (\mathbf{X}, \mathbf{y}) = \{(x_1, y_1), \ldots, (x_n, y_n)\}$ where $y_i = f(x_i) + \epsilon_i$ are noisy observations of the true function with Gaussian noise $\epsilon_i \sim \mathcal{N}(0, \lambda_i)$. Then, when performing Bayesian inference, we can express the posterior mean and covariance of the GP as

$$\mu(x) = \mu_{\text{prior}}(x) + k(x, \mathbf{X})\big(k(\mathbf{X}) + \text{diag}(\lambda)\big)^{-1}(\mathbf{y} - \mu_{\text{prior}}(x)) , \qquad (6)$$

$$\sigma^2(x) = k(x, x) - k(x, \mathbf{X})\big(k(\mathbf{X}) + \text{diag}(\lambda)\big)^{-1}k(\mathbf{X}, x) . \qquad (7)$$

In practice, the prior mean and kernel may have hyperparameters $\theta$ which specify what functions it is able to model. For example, many kernel functions include lengthscale values which govern how correlated the function output is when a certain input dimension changes. One method to find the optimal hyperparameters for the kernel is by finding the hyperparameter which maximizes the marginal log-likelihood.

In Bayesian optimization (BO), the goal is to find the maxima of the unknown function $f$. This function is black-box, and assumed to have no analytical form. To do so, we can learn more about

$f$ by querying it at different inputs, and perform Bayesian inference to update our belief on the unknown function.

Given the current observations $\mathcal{D}_t = (\mathbf{X}_t, \mathbf{y}_t) = \{(x_1, y_1), \ldots, (x_t, y_t)\}$ in round $t$ of data selection, GP regression can be performed to obtain a posterior mean $\mu_t$ and posterior variance $\sigma_t^2$. The next input to query $x_{t+1}$ can be chosen as the input which maximizes some acquisition function. Examples of such acquisition function include the expected improvement (Jones et al., 1998)

$$\text{EI}_t(x) = \mathbb{E}_{y' \sim \mathcal{N}(\mu_{t-1}(x), \sigma_{t-1}^2(x))} \Big[ \max(0, y' - \max_{y \in \mathbf{y}_{t-1}} y) \Big] \tag{8}$$

or the upper confidence bound (Srinivas et al., 2012)

$$\text{UCB}_t(x) = \mu_{t-1}(x) + \beta_t \sigma_{t-1}(x) \tag{9}$$

where $\beta_t > 0$ is a constant that may vary with $t$. In all of these acquisition functions, a tradeoff is performed between selecting inputs that the GP is uncertain about (i.e., with high $\sigma_{t-1}^2(x)$) to learn more about those unknown region, and selecting inputs in regions where the function value is known to be higher (i.e., with high $\mu_{t-1}(x)$).

## C    DETAILED DISCUSSION ON THE PROBLEM SETUP

### C.1    HYPERPARAMETERS CONSIDERED

in Table 1, we list several hyperparameters which we include in our parallelism configuration and the range of the values. Note that the hyperparameters are constrained to give a valid PC as well; for example, we ensure that $\texttt{dp} \cdot \texttt{tp} \cdot \texttt{pp} = \texttt{n\_gpus}$ to ensure that each dimension do not exceed number of GPUs. Some hyperparameters are also set to their default value when not in use; for example, if $\texttt{pp} = 1$ (i.e., no PP used) then we restrict $\texttt{mb} = \texttt{mc} = 1$ such that PCs are not duplicated. In the code, we generate all possible PCs beforehand so we can ensure that all PCs chosen will be valid according to the constraints.

Table 1: Tunable hyperparameters in a parallelism configuration.

| Hyperparameter | Description | Feasible Values |
|---|---|---|
| DP size (dp) | Data parallelism degree | [1, n_gpus] |
| TP size (tp) | Tensor parallelism degree | [1, n_gpus] |
| PP size (pp) | Pipeline parallelism degree | [1, n_gpus] |
| DP bucket size | Size for gradient reduction buckets (MB) | [1, 4096] |
| ZeRO stage | ZeRO stage used | [0, 3] |
| ZeRO bucket size | Bucket size for ZeRO communication | [1, 4096] |
| Overlap ZeRO communication | Whether to overlap ZeRO communication | True / False |
| Overlap ZeRO AllGather | Whether to overlap AllGather | True / False |
| # microbatches (mb) | Number of microbatches per forward pass | ≤ batch size |
| # model chunks (mc) | Number of model chunks for pipelining | ≤ # transformer blocks |
| Overlap P2P for PP | Overlap PP communication or not | True / False |
| Grad. checkpointing | Whether gradient checkpointing is enabled | True / False |

### C.2    THROUGHPUT VERSUS TIME PER TRAINING STEP

We explain why we choose to maximize throughput instead of minimizing time per training step.

As an example, suppose we consider three PCs $H, H', H''$ where the times per training step are given by $\mathcal{T}(H) = 0.3$, $\mathcal{T}(H') = 0.4$, and $\mathcal{T}(H'') = 0.5$. In this case, $H$ would be the best PC out of the three. We see here the gap of the time per training step between $H$ and $H'$ is $\mathcal{T}(H') - \mathcal{T}(H) = 0.1$, and the same gap size for $H'$ and $H''$ of $\mathcal{T}(H'') - \mathcal{T}(H') = 0.1$. Meanwhile, the gap between the throughput of the two PCs would be $\mathcal{R}(H) - \mathcal{R}(H') = \mathcal{T}(H)^{-1} - \mathcal{T}(H')^{-1} = 3.\overline{3} - 2.5 = 0.8\overline{3}$ and $\mathcal{R}(H') - \mathcal{R}(H'') = \mathcal{T}(H)^{-1} - \mathcal{T}(H')^{-1} = 2.5 - 2 = 0.5$. We can see that the gap between the best PC becomes enhanced when we consider the throughput, when we compare it with the relative gap size of the training step time.

More concretely, if we have a PC which requires time $t$ per training step, then you can reduce it by an amount of $\Delta t$, then the throughput would have increased by an amount $(\Delta t)/t^2$. When $t$ becomes

smaller, the change in throughput will also increase but at an increasing rate. This therefore means by modeling the throughput, the scores of the good PCs will be more clearly separated.

Additionally, we also consider a maximization of throughput since throughput would be bounded by $[0, r_{\max}]$, rather than the time per training step which would be unbounded on one end, i.e., be in the interval $[t_{\min}, \infty)$, which makes suboptimal PCs easier to handle. Also, we frame our problem as a maximization in order to be consistent with Bayesian optimization works which typically considers a maximization problem.

# D   DETAILED DISCUSSION OF GP SURROGATE IN OPPA

## D.1   PARALLELISM-INFORMED PRIOR MEAN FOR THE THROUGHPUT

In this section, we elaborate on how the throughput prior mean is constructed in order to obtain the form for the parallelism-informed prior mean. In summary, we design the prior to incorporate the following characteristics.

- *Parallelism coverage.* We model DP/TP traffic via All-Reduce–style collectives and PP via point-to-point transfers, using a placement-aware split of intra- and inter-host links. For computation costs, we also explicitly consider the pipeline bubbles.

- *Topology and protocol awareness.* The communication term uses canonical ring/tree scaling with hierarchical aggregation (node-local first, then cross-node), while collapsing ZERO stages into a single All-Reduce surrogate at the bandwidth level (bytes over the wire are of the same order) and letting stage-specific latency/overlap differences be absorbed by $\mathbf{C}$ (e.g., event counts, bucket sizes, communication overhead for each parallelism dimension, etc.).

- *Hardware agnosticism via learning.* Rather than hard-coding device/network constants, we expose a small set of effective coefficients that are learned from a few traces. This keeps the prior portable across models, data types, and interconnects while preserving the correct asymptotic trends for dp, tp, and pp.

To estimate the computation time, we assume an idealized machine that allows infinite parallelization, such that DP and TP are perfectly parallelized. Meanwhile, PP using an interleaved schedule incurs additional computation time from the microbatches being ran sequentially, and from pipeline bubble when the first microbatch is being fed through the pipeline (Narayanan et al., 2019). This additional computation time from PP, visualized in Fig. 9, is roughly equal to

$$\hat{\mathcal{T}}_{\text{comp}}(H; t_{\text{comp}}) = \frac{t_{\text{comp}}}{\text{n\_gpus}} \cdot \left( \text{mb} + \frac{\text{pp} - 1}{\text{mc}} \right) \tag{10}$$

where mb is the number of microbatches used in PP (set to 1 when PP is not used), mc is the number of model chunks for PP (also set to 1 when PP is not used), and $t_c = t_f + t_b$ is the total time to perform the forward and backward passes.

To estimate the communication time, inspired by the model visualized in Fig. 2, we assume that DP and TP involve All-Reduce communications, and PP P2P communications, where these costs are modeled separately. We use an extended $(\alpha, \beta, \gamma)$ model dicussed by Xiong et al. (2024), and for the intra- and inter-host communications, we characterize the network performance by the latency $\alpha_{\text{intra}}$ and $\alpha_{\text{inter}}$, the per-byte bandwidth cost $\beta_{\text{intra}}$ and $\beta_{\text{inter}}$, the incast overhead $\gamma_{\text{intra}}$ and $\gamma_{\text{inter}}$, and the memory access overhead $\delta_{\text{intra}}$ and $\delta_{\text{inter}}$. We assume that the inter-host communication costs will be larger than their intra-host counterparts.

For DP and TP, we assume that the Ring All-Reduce implementation is used, where the cost to gather and scatter data of size $N$ across $D$ machines is given by

$$C_{\text{AR}}(D, N, \alpha, \beta, \gamma, \delta) = 2(D - 1)\alpha + \frac{D - 1}{D} N(2\beta + \gamma + 3\delta). \tag{11}$$

For DP, we consider the gradient synchronization from the All-Reduce algorithm. The data size per GPU for DP All-Reduce, $N_{\text{dp}}$, is

$$N_{\text{dp}} = \lambda_Z \frac{M_{\text{model}}}{\text{tp} \cdot \text{pp}} \tag{12}$$

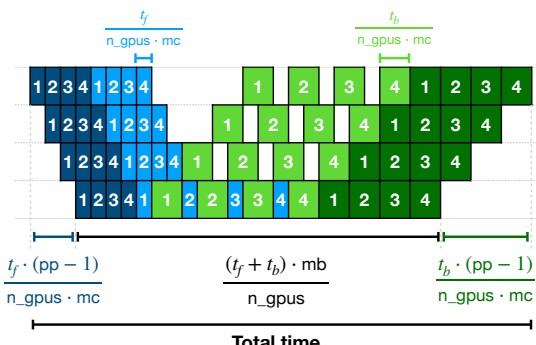

Figure 9: Estimate for computation time for PP where $t_f$ and $t_b$ are the time required for the forward and backward stages respectively, for when $\mathrm{pp} = \mathrm{n\_gpus} = 4$, $\mathrm{mb} = 4$, and $\mathrm{mc} = 2$

where $M_{\mathrm{model}}$ is a learnable total model parameter size, and $\lambda_Z$ is a tiny fudge for ZERO flavor while staying in AR-land, and it accounts for param-All-Gather + grad-Reduce-Scatter volume equivalence with small overhead for ZERO-3.

Suppose $G_{\mathrm{node}}$ is the number of GPUs per node. The overall communication cost from DP $\hat{\mathcal{T}}_{\mathrm{comm,dp}}$ would then depend on the configuration of the network as follows:

- In a **hierarchical** (multi-host scenario with $\mathrm{dp} > G_{\mathrm{node}}$) system, the cost is given by

$$\hat{\mathcal{T}}_{\mathrm{comm,dp}}(H; \mathbf{C}) = C_{\mathrm{AR}}(G_{\mathrm{node}}, N_{\mathrm{dp}}, \alpha_{\mathrm{intra}}, \beta_{\mathrm{intra}}, \gamma_{\mathrm{intra}}, \delta_{\mathrm{intra}})$$
$$+ C_{\mathrm{AR}}(\lceil \mathrm{dp}/G_{\mathrm{node}} \rceil, N_{\mathrm{dp}}/G_{\mathrm{node}}, \alpha_{\mathrm{inter}}, \beta_{\mathrm{inter}}, \gamma_{\mathrm{inter}}, \delta_{\mathrm{inter}}). \quad (13)$$

- In a **flat inter-node** (multi-host scenario with $\mathrm{dp} \leq G_{\mathrm{node}}$), the cost is given by

$$\hat{\mathcal{T}}_{\mathrm{comm,dp}}(H; \mathbf{C}) = C_{\mathrm{AR}}(\mathrm{dp}, N_{\mathrm{dp}}, \alpha_{\mathrm{inter}}, \beta_{\mathrm{inter}}, \gamma_{\mathrm{inter}}, \delta_{\mathrm{inter}}). \quad (14)$$

- In a **flat intra-node** (single-host scenario), the cost is given by

$$\hat{\mathcal{T}}_{\mathrm{comm,dp}}(H; \mathbf{C}) = C_{\mathrm{AR}}(\mathrm{dp}, N_{\mathrm{dp}}, \alpha_{\mathrm{intra}}, \beta_{\mathrm{intra}}, \gamma_{\mathrm{intra}}, \delta_{\mathrm{intra}}). \quad (15)$$

Depending on the hardware used, the appropriate cost for the scenario can be selected.

For TP, we consider the cost from frequent activation communication (e.g., All-Reduce per layer). Let $M_{\mathrm{act,tp}}$ be a learnable characteristic data size for one such TP All-Reduce operation. Let $O_{\mathrm{tp/mb}}$ be the learnable number of these operations per microbatch. The total number of TP communication operations is $N_{\mathrm{tp,ops}} = O_{\mathrm{tp/mb}} \cdot \mathrm{mb}$, and the cost of a single TP All-Reduce operation is $C_{\mathrm{AR}}(\mathrm{tp}, M_{\mathrm{act,tp}}, \alpha_{\mathrm{eff}}, \beta_{\mathrm{eff}}, \gamma_{\mathrm{eff}}, \delta_{\mathrm{eff}})$ where effective parameters $\alpha_{\mathrm{eff}}, \beta_{\mathrm{eff}}, \gamma_{\mathrm{eff}}, \delta_{\mathrm{eff}}$ are chosen as intra-node or inter-node based on whether the $\mathrm{tp}$ group spans multiple hosts (i.e., if $N_H > 1$ and $\mathrm{tp} > G_{\mathrm{node}}$) or not. Then, the total TP communication cost is the number of communication operations multiplied by the cost per communication operation, or

$$\hat{\mathcal{T}}_{\mathrm{comm,tp}}(H; \mathbf{C}) = N_{\mathrm{tp,ops}} \cdot C_{\mathrm{AR}}(\mathrm{tp}, M_{\mathrm{act,tp}}, \alpha_{\mathrm{eff}}, \beta_{\mathrm{eff}}, \gamma_{\mathrm{eff}}, \delta_{\mathrm{eff}}). \quad (16)$$

For PP, we consider the point-to-point (P2P) transfers of activations and gradients between $\mathrm{pp}$ pipeline stages, where the cost to transfer data of size $N$ is given by

$$C_{\mathrm{P2P}}(N, \alpha, \beta) = \alpha + N \cdot \beta. \quad (17)$$

Let $M_{\mathrm{act,pp}}$ be a learnable characteristic data size for one P2P transfer (e.g., the size of an activation tensor). The number of communication boundaries is $\mathrm{pp} - 1$. Communication occurs for each of $\mathrm{mb}$ microbatch, in both forward (activations) and backward (gradients) directions. The total number of P2P communications is given by $N_{\mathrm{pp,transfers}} = \max(\mathrm{pp} - 1, 0) \cdot 2\mathrm{mb}$, where a single P2P transfer uses effective latency $\alpha_{\mathrm{eff}}$ and effective per-unit-data cost $\beta_{\mathrm{pp,eff}}$ (derived from base $\beta$ parameters), chosen as intra-node or inter-node based on whether communicating stages are on different hosts

(approximated if $N_H > 1$). Given this, the overall cost of all communications related to PP would be given by

$$\hat{\mathcal{T}}_{\text{comm,pp}}(H; \mathbf{C}) = N_{\text{pp,transfers}} \cdot C_{\text{P2P}}(M_{\text{act,pp}}, \alpha_{\text{eff}}, \beta_{\text{pp,eff}}). \tag{18}$$

Here, if $\texttt{pp} = 1$, then $\hat{\mathcal{T}}_{\text{comm,pp}} = 0$.

When combining the communication costs for all three types of parallelism, considering in practice large chunks of communication overlap with compute, we attach non-overlap factors $\kappa$ and obtain

$$\hat{\mathcal{T}}_{\text{comm}}(H; \mathbf{C}) = \kappa_{\text{dp}} \cdot \hat{\mathcal{T}}_{\text{comm,dp}}(H; \mathbf{C}) + \kappa_{\text{tp}} \cdot \hat{\mathcal{T}}_{\text{comm,tp}}(H; \mathbf{C}) + \kappa_{\text{pp}} \cdot \hat{\mathcal{T}}_{\text{comm,pp}}(H; \mathbf{C}) \tag{19}$$

where $\mathbf{C}$ are constants related to the various costs which are to be inferred. Given Eqs. (10) and (19), we can construct the prior mean for throughput to be

$$\hat{\mathcal{R}}(H; \{t_{\text{comp}}, \mathbf{C}\}) = \left[ \hat{\mathcal{T}}_{\text{comp}}(H; t_{\text{c}}) + \hat{\mathcal{T}}_{\text{comm}}(H; \mathbf{C}) \right]^{-1}. \tag{20}$$

### D.2 PARALLELISM-INFORMED PRIOR MEAN FOR THE MAXIMUM MEMORY USAGE

We briefly elaborate on the choice of prior mean in Eq. (21). As discussed, we only consider the memory that is required to store the NN parameters, and those to compute the gradient updates.

For NN parameters, its sharding can be done on the pipeline or on the layers, allowing us to approximate the GPU memory required for storing the NN parameters to be inversely proportional to $\texttt{pp} \cdot \texttt{tp}$. Note that assuming the simplest DP implementation, the NN parameters are duplicated and stored on each DP dimension, and so the maximum memory usage is not affected by the DP.

Meanwhile, in the case of backpropagation computation, the maximum memory used will roughly be proportional to how many model parameters a certain GPU has to perform the forward and backward passes for, times how many training samples the GPU has to process at any one time. We expect this quantity to be inversely proportional to the number of total GPUs times the latter to depend on the number of microbatches used.

Combining these two factors, can write the maximum memory usage as

$$\hat{\mathcal{M}}(H; \theta_{\mathcal{M}}) = \min \left\{ m_1 \cdot (\texttt{pp} \cdot \texttt{tp})^{-1} + m_2 \cdot (\texttt{n\_gpus} \cdot \texttt{mb})^{-1} + m_3, \ M_0 \right\} \tag{21}$$

where, $m_1$ captures the memory used for storing model parameters, and $m_2$ captures the memory used during backpropagation computations, $m_3$ are any other additional memory overheads unaccounted for by our model, and $\theta_{\mathcal{M}} = \{m_1, m_2, m_3\}$. Note that since we cannot measure maximum memory usage above values of $M_0$, we apply the $\min$ function to clip the prior belief function.

### D.3 KERNEL

Given the embedding $e(H)$, we use the Matern kernel (Rasmussen & Williams, 2006), which is given by

$$k(H, H') = \sigma_k^2 k_{\text{Matern},\nu}\big(e(H), e(H'); \ell\big) = \sigma_k^2 \frac{2^{1-\nu}}{\Gamma(\nu)} \big(\sqrt{2\nu}\, d_\ell(H, H')\big)^\nu K_\nu\big(\sqrt{2\nu}\, d_\ell(H, H')\big)$$

where $\Gamma$ is the Gamma function, $K_\nu$ is the modified Bessel function, $\sigma_k$ is the kernel scaling constant,

$$d_\ell(H, H') = \big(e(H) - e(H')\big)^\top \mathbf{L}^{-2} \big(e(H) - e(H')\big) \tag{22}$$

is the distance between two PC embeddings and $\mathbf{L} = \text{diag}(\ell) = \text{diag}([\, \ell_1 \cdots \ell_p \,])$ is the lengthscale.

### D.4 POSTERIOR DISTRIBUTION OF PREDICTED THROUGHPUT AND MEMORY USAGE

Suppose we are in the $i$th round. We let $\mathbf{H}_i = [H_1 \cdots H_i]$ be the list of PCs, $\bar{\mathbf{r}}_i = [\bar{r}_{1,\hat{q}_1} \cdots \bar{r}_{i,\hat{q}_i}]$ and $\sigma_{\bar{\mathbf{r}}_i}^2 = [\sigma_{\bar{r}_{1,\hat{q}_1}}^2 \cdots \sigma_{\bar{r}_{i,\hat{q}_i}}^2]$ be the observed throughput and the corresponding variance, and $\mathbf{m}_i = [m_1 \cdots m_i]$ be the observed maximum memory usage values.

Given the data, we first find the optimal hyperaparameters $\theta = \{\theta_{\mathcal{R}}, \theta_{\mathcal{M}}, \theta_k\}$. This is done by maximizing the marginal log-likelihood (Rasmussen & Williams, 2006), or

$$\theta = \underset{\theta'}{\arg\max} \ \log p(\bar{\mathbf{r}}_i, \bar{\mathbf{m}}_i | \bar{\mathbf{H}}_i, \theta). \tag{23}$$

The posterior mean and variance for the throughput for some PC $H'$ can then be defined as

$$\mu_{\mathcal{R},i}(H') = \hat{\mathcal{R}}(\mathbf{H}_i; \theta_{\mathcal{R}}) + k(H', \mathbf{H}_i; \theta_k)\big(k(\mathbf{H}_i, \mathbf{H}_i; \theta_k) + \text{diag}(\sigma_{\bar{\mathbf{r}}_i}^2)\big)^{-1}\big(\bar{\mathbf{r}}_i - \hat{\mathcal{R}}(\mathbf{H}_i; \theta_{\mathcal{R}})\big), \quad (24)$$

$$\sigma_{\mathcal{R},i}^2(H') = k(H', H'; \theta_k) - k(H', \mathbf{H}_i; \theta_k)\big(k(\mathbf{H}_i, \mathbf{H}_i; \theta_k) + \text{diag}(\sigma_{\bar{\mathbf{r}}_i}^2)\big)^{-1}k(\mathbf{H}_i, H'; \theta_k), \quad (25)$$

and the posterior mean and variance for the memory usage for some PC $H'$ can then be defined as

$$\mu_{\mathcal{M},i}(H') = \hat{\mathcal{M}}(\mathbf{H}_i; \theta_{\mathcal{M}}) + k(H', \mathbf{H}_i; \theta_k)\big(k(\mathbf{H}_i, \mathbf{H}_i; \theta_k) + \lambda I\big)^{-1}\big(\mathbf{m}_i - \hat{\mathcal{M}}(\mathbf{H}_i; \theta_{\mathcal{M}})\big), \quad (26)$$

$$\sigma_{\mathcal{M},i}^2(H') = k(H', H'; \theta_k) - k(H', \mathbf{H}_i; \theta_k)\big(k(\mathbf{H}_i, \mathbf{H}_i; \theta_k) + \lambda I\big)^{-1}k(\mathbf{H}_i, H'; \theta_k) \quad (27)$$

where $\lambda$ is to make the matrix invertible.

# E    DETAILED DISCUSSION OF PC SELECTION METHOD IN OPPA

## E.1    RANDOM SAMPLING FOR ADDITIONAL EXPLORATION

In OPPA, we sometimes select PCs at random for additional exploration. There are two scenarios which triggers a random selection of PC in OPPA.

1. In the first few chosen PCs. This is because in the beginning there are no PCs which can be used to infer the hyperparameters for the prior distribution of the GP, therefore a few PCs are chosen at random to kick-off the modeling process and provide a reasonably diverse set of samples to infer the hyperparameters well.

2. When too many out-of-memory errors have been encountered in a row. This is because any out-of-memory trials will not result in a usable training data for the throughput modeling and possibly minimal data for the maximum memory GP, which does not aid the GP model. When too many such cases are encountered, we attempt to do random exploration so that the model can receive some information that can be used to model better with and find new feasible PCs.

For the random selection process, we select a PC using a weighted random strategy, such that the probability of obtaining a PC with a certain parallelism dimension size configurations are equal.

# F    DETAILED DISCUSSION PC TRIALING METHOD

## F.1    THROUGHPUT ESTIMATION

Given the time $t_{i,1}, \ldots, t_{i,q}$ required for $q$ training steps, we can estimate the throughput and its predicted variance as

$$\bar{r}_{i,q} = \frac{1}{q}\sum_{j=1}^{q}\frac{1}{t_{i,j}}, \quad \text{and} \quad \sigma_{\bar{r}_{i,q}}^2 = \frac{1}{q}\sum_{j=1}^{q}\left(\frac{1}{t_{i,j}} - \bar{r}_{i,q}\right)^2. \quad (28)$$

## F.2    OUTLIER REMOVAL

As demonstrated in Fig. 3, not all training time measurements will be representative of the true throughput. We therefore perform two actions. First, we remove the first training step, $t_{i,1}$, since it typically corresponds to a warm-up for the training and therefore will usually be an anomaly measurement. Second, we compute the median and the inter-quartile range ($IQR$), and remove all measurements which are away from the median by at least $2 \times IQR$. This is a standard method for outlier removal in practice. However, we choose a looser threshold for outliers to account for fluctuating measurements. This generally has little effect on the overall optimization since most training steps tend to not fluctuate by too much anyway. This is demonstrated by Fig. 10 which shows that the predicted throughput would have little difference across the threshold of outlier removal, showing robustness of our method to the outlier removal process. The remaining training points are then used to compute Eq. (28).

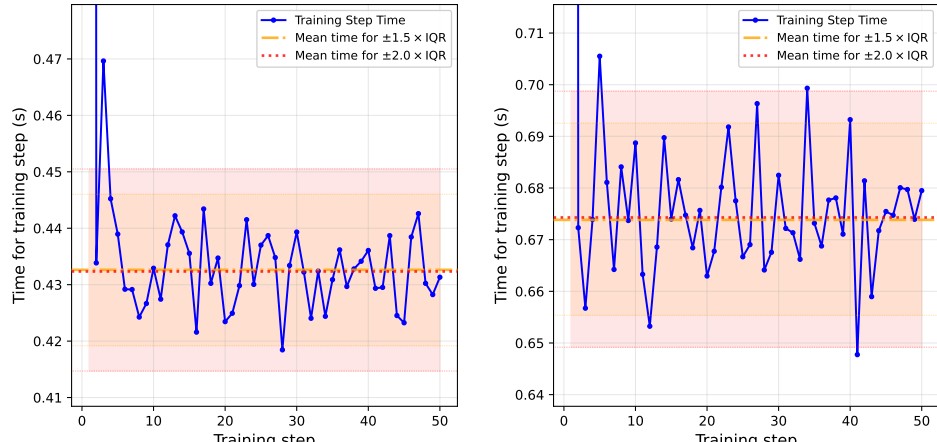

Figure 10: Example of the outlier removal process based on the IQR on different random PC trials. Here, we show the effect of the threshold selected versus the mean of the measurements, demonstrating the robustness with respect to the different thresholds.

### F.3 ADDITIONAL DISCUSSION ON EARLY TRIAL TERMINATION MECHANISM

In this section, we attempt to prove Thm. 4.1. To do so, we will consider a more general case for an arbitrary unknown function $f$. We will first provide an intuitive justification for why early termination can be applied, followed by a theoretical proof for Thm. 4.1 and ablation studies on synthetic functions.

#### F.3.1 INTUITION BEHIND EARLY TERMINATION MECHANISM

Before stating the formal proof, we provide some intuition behind why the early termination mechanism saves on resources while also not affecting the overall performance. In Fig. 11, we present a toy example of a BO iteration where an input is repeatedly queried.

Here, we notice several things. We first see that for a certain input, more repetitions of the query will result in a lower standard deviation for the overall observation. This is simply from the fact that when we take the mean of several i.i.d. observations, the mean will tend towards the true value, and the variance of the mean predictor will decay at a rate inversely proportional to the number of observations. This is reflected in the plots where the observation at $x = 2$ has smaller and smaller standard deviation as we have more repeats, and as we will show in Corollary F.2. In these cases, we see that some repeated queries are therefore necessary to reduce the noise of the observations.

However, as we perform more repeats, there is a diminishing effect on how much more information is gained from each additional repeats. In the diagram, we see that after 20 repeats of the observations (as in Fig. 11c), the GP would barely change with more repeats of the same observation. Similarly, we see that at 20 repeats, we are able to somewhat conclude that the global optimum is less likely to be in the neighborhood of $x = 2$, as reflected by the corresponding probability density function, and that this belief remains when we continue up to 100 repeats (as in Fig. 11d). This shows the additional repeats are unlikely to improve our surrogate prediction and the confidence of the optimal. We therefore can see that by stopping after 20 repeats, we would have been able to eventually make the same conclusion about the optimum in the end while saving on resources that would have been incurred from repeated queries.

Finally, we note that additional repeats are more beneficial to recovering the optimum value in the case that it improves upon the best observation so far. In Fig. 12, we see that additional repeats of observation at $x = 2$ eventually results in diminishing returns, as seen in the probability distribution of the maximum value of $f(x)$ being similar between using 20 repeats and 100 repeats. Meanwhile, in Fig. 13, where we instead repeatedly query the input $x = 4.2$, we continue to gain information about the maximum value even when we use 100 repeats, as seen by the sharper probability distribution for

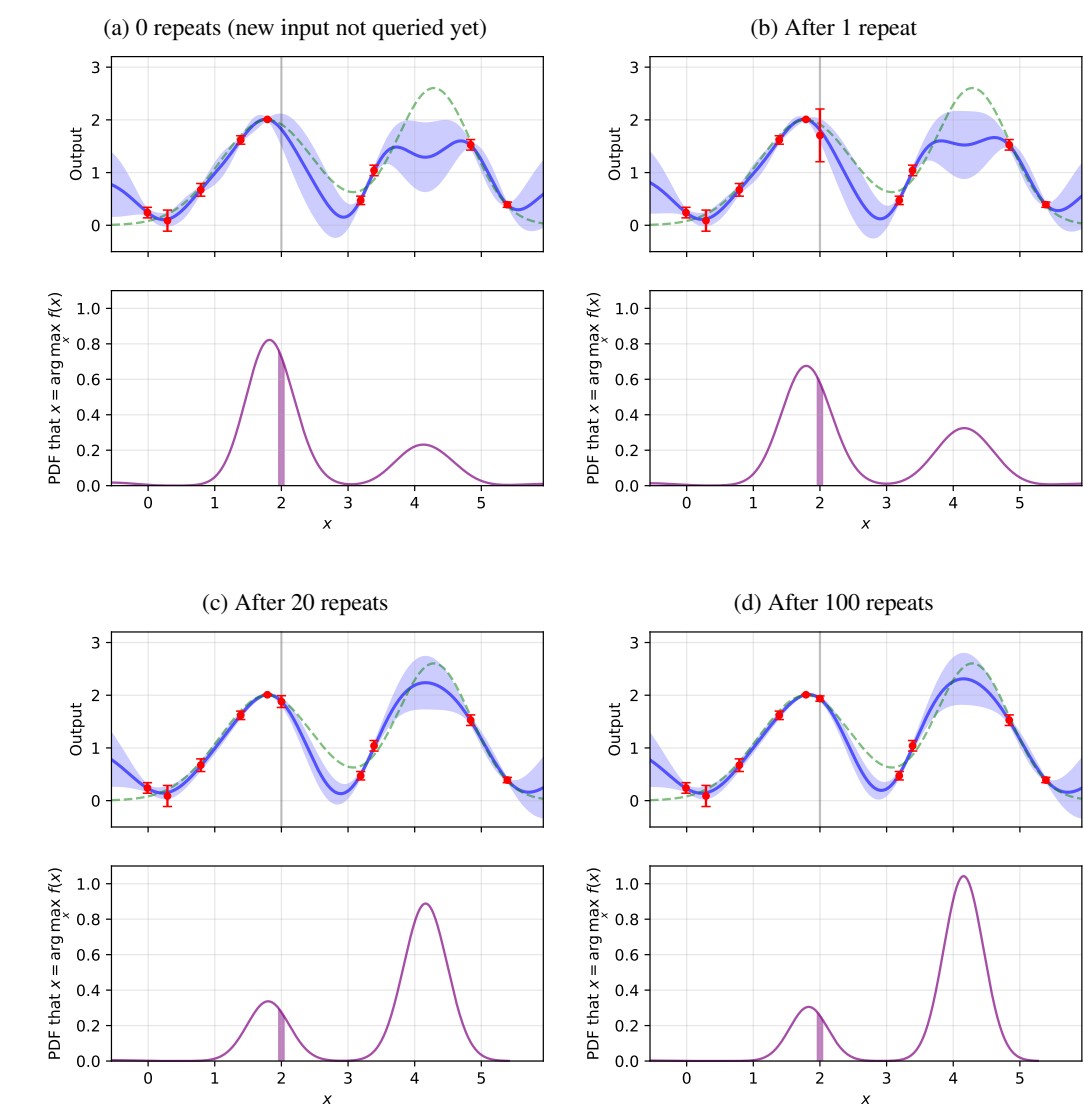

Figure 11: Demonstration of why early termination of certain trials does not affect final result. Figs. 11a to 11d represent the modeled GP and the predicted optimum when a certain input, in this case at $x = \arg\max_{x'} \mu_t(x') + \beta_t \sigma_t(x') = 2$, is repeatedly queried for 0, 1, 20, and 100 times. For each subfigure, the top diagram represents the modeled GP, where the dashed green line represents the true function, the blue line and error band represent the GP mean and standard deviation, and the red points represent the observation mean and standard deviation (the latter which shrinks with more repeats). In the bottom diagram, we show the probability of each input being the optimal (i.e., probability that $x = \arg\max_{x'} f(x')$), while highlighting the interval around the input $x = 2$.

the belief of the maximum $f(x)$. We therefore can see that additional repeats are less informative if they are for suboptimal inputs. We can therefore use this idea to form a metric to determine if a trial should be terminated early.

### F.3.2 PROOF OF THM. 4.1

With this intuition, we will now formally prove the performance of the early termination mechanism. For completeness, we first state the assumptions for the function and the observations, which follows from other BO works (Srinivas et al., 2012; Makarova et al., 2021; Kirschner & Krause, 2018) however with additional assumptions on repeated observations from the same input.

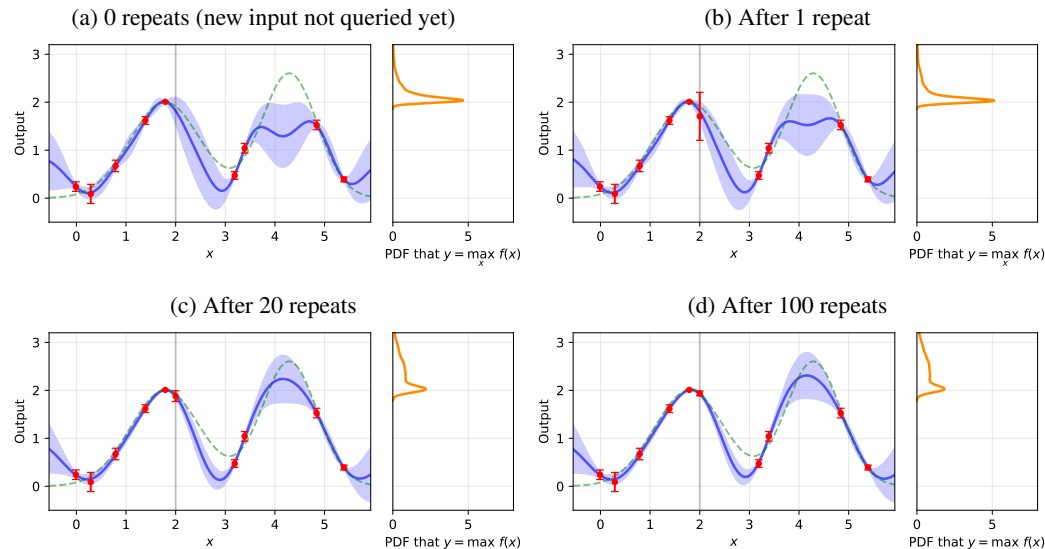

Figure 12: Additional demonstration of why early termination of certain trials does not affect the obtained optimal value. The setup is as done previously in Fig. 11, with queries at $x = 2$. For each subfigure, the left plot represents the modeled GP as shown previously in Fig. 11. In the right diagram, we show the probability density function for the belief of the maximum value $\max_x f(x)$ according to the GP.

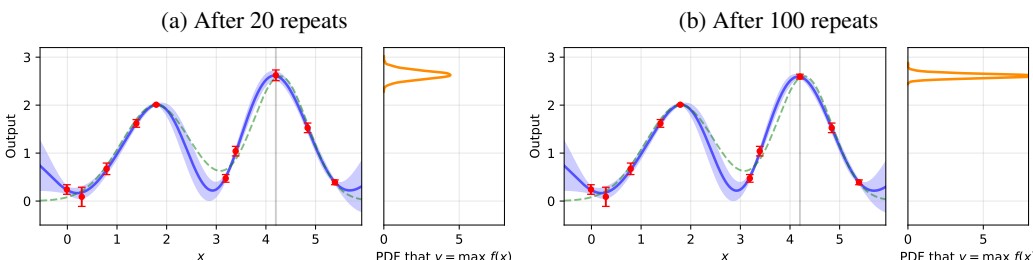

Figure 13: Additional demonstration of why early termination of certain trials does not affect the obtained optimal value. The setup is similar to as done previously in Fig. 11, however instead with queries at $x = 4.2$, and with only 20 repeats and 100 repeats shown (comparable to Figs. 12c and 12d). The information in each subfigure is the same as in Fig. 12.

**Assumption F.1.** Let $f \sim \mathcal{GP}(0, k)$ be an unknown function drawn from a GP with zero mean and kernel function $k$, where the RKHS norm $\|f\|_{\mathcal{H}} \leq B$ is bounded. The BO procedure is as noted in Algorithm 2, where in each BO iteration $i$, an input $x_i \in \mathcal{X}$ is selected, and $\hat{q}_i \leq q_{\max}$ noisy outputs $y_{i,j} = f(x_i) + \varepsilon_{i,j}$ are returned, where $\varepsilon_{i,j} \sim \mathcal{N}(0, s^2)$ are i.i.d. noise.

We now also show Algorithm 1 which repeats each query $q_{\max}$ times, and our proposed Algorithm 2 which does early termination on some of the rounds. Our theoretical results in this section will consider Algorithm 2.

We first state a result regarding the mean estimator of $f(x_i)$.

**Corollary F.2.** *Suppose we define*

$$\bar{y}_{i,q} = \frac{1}{q} \sum_{j=1}^{q} y_{i,j}. \tag{29}$$

*Then, the expected value of $\bar{x}_i$ is $\mathbb{E}[\bar{y}_{i,q}] = f(x_i)$, and its variance bounded by $\mathbb{V}[\bar{y}_{i,q}] = s^2/q$.*

---

**Algorithm 1** GP-UCB with Repeated Trials

---

1: $\mathcal{D}_0 \leftarrow \emptyset$
2: **for** $i = 1, \ldots, N$ **do**
3:     Fit GP on $\mathcal{D}_{i-1}$ to obtain mean $\mu_{i-1}$ and variance $\sigma_{i-1}^2$
4:     $x_i \leftarrow \arg\max_{x \in \mathcal{X}} \mu_{i-1}(x) + \beta_i \sigma_{i-1}(x)$
5:     **for** $j = 1, \ldots, q_{\max}$ **do**
6:        Sample $y_{i,j} = f(x_i) + \varepsilon_{i,j}$
7:     **end for**
8:     $\bar{y}_{i,q_{\max}} \leftarrow q_{\max}^{-1} \sum_{j=1}^{q_{\max}} y_{i,j}$
9:     $\mathcal{D}_i \leftarrow \mathcal{D}_i \cup \{(x_i, \bar{y}_{i,q_{\max}})\}$
10: **end for**

---

**Algorithm 2** Modified GP-UCB with Early Trial Termination

---

1: $\mathcal{D}_0 \leftarrow \emptyset$
2: **for** $i = 1, \ldots, N$ **do**
3:     Fit GP on $\mathcal{D}_{i-1}$ to obtain mean $\mu_{i-1}$ and variance $\sigma_{i-1}^2$
4:     $x_i \leftarrow \arg\max_{x \in \mathcal{X}} \mu_{i-1}(x) + \beta_i \sigma_{i-1}(x)$
5:     **for** $j = 1, \ldots, q_{\max}$ **do**
6:        Sample $y_{i,j} = f(x_i) + \varepsilon_{i,j}$
7:        $\bar{y}_{i,j} \leftarrow j^{-1} \sum_{j'-1}^{j} y_{i,j'}$
8:        $\mathcal{D}'_{i,j'} \leftarrow \mathcal{D}_i \cup \{(x_i, \bar{y}_{i,j})\}$
9:        **if** $\bar{y}_{i,q} < \max_{j<i} \bar{y}_{j,\hat{q}_j} + \tau_q$ **then**
10:           **break**
11:        **end if**
12:     **end for**
13:     $\mathcal{D}_i \leftarrow \mathcal{D}'_{i,j'}$
14: **end for**

---

*Proof.* The results are direct consequences of the summation of expected values and variances of independent random variables, and the fact that $\mathbb{V}[y_{i,j}] = s^2$ by assumption. □

We will now prove the first half of Thm. 4.1. Given the results in Corollary F.2, we next show that we are able to obtain a GP with good estimate bounds even if some trials are terminated early, i.e., not ran for up to $q_{\max}$ repeats.

**Lemma F.3.** *Suppose we let*

$$\mu_i(x) = k(x, \mathbf{X}_i)\big(k(\mathbf{X}_i, \mathbf{X}_i) + s^2 \mathbf{Q}_i^{-1}\big)^{-1} \mathbf{y}_i \,, \tag{30}$$

$$\sigma_i^2(x) = k(x, x) - k(x, \mathbf{X}_i)\big(k(\mathbf{X}_i, \mathbf{X}_i) + s^2 \mathbf{Q}_i^{-1}\big)^{-1} k(\mathbf{X}_i, x) \tag{31}$$

*where* $\mathbf{X}_i = [x_1 \cdots x_i]$, $\mathbf{y}_i = [\bar{y}_{1,\hat{q}_1} \cdots \bar{y}_{i,\hat{q}_i}]$, *and* $\mathbf{Q}_i = \mathrm{diag}([\hat{q}_1 \cdots \hat{q}_i])$. *If*

$$\beta_i = B + \sqrt{2 \log \frac{\det\big(k(\mathbf{X}_i, \mathbf{X}_i) + s^2 \mathbf{Q}_i^{-1}\big)^{1/2}}{\delta \det\big(s^2 \mathbf{Q}_i^{-1}\big)^{1/2}}} \tag{32}$$

*then, with probability greater than* $1 - \delta$*, for all* $x \in \mathcal{X}$ *and all* $i = 1, \ldots, N$*, we have*

$$\big|f(x) - \mu_{i-1}(x)\big| \leq \beta_i \sigma_{i-1}(x). \tag{33}$$

*Proof.* Given the variance of the mean predictor from Corollary F.2, our scenario can be thought of as having $i$ observations with heteroscedastic noise with variances of $s^2/\hat{q}_1, \ldots, s^2/\hat{q}_i$. The variance bounds then follow directly from Lemma 7 in Kirschner & Krause (2018) where we substitute $\Sigma_i \to s^2 \mathbf{Q}_i^{-1}$ and $\lambda \to 1$. □

**Corollary F.4.** *Given Lemma F.3, for all* $x' \in \mathcal{X}$*, we have* $f(x') - \mu_{i-1}(x_i) \leq \beta_i \sigma_{i-1}(x_i)$.

*Proof.* We see that

$$f(x') - \mu_{i-1}(x_i) \le \mu_{i-1}(x') + \beta_i \sigma_{i-1}(x') - \mu_{i-1}(x_i) \qquad \text{by Lemma F.3,} \qquad (34)$$

$$\le \mu_{i-1}(x_i) + \beta_i \sigma_{i-1}(x_i) - \mu_{i-1}(x_i) \qquad \text{by how } x_i \text{ is chosen,} \qquad (35)$$

$$= \beta_i \sigma_{i-1}(x_i). \qquad (36)$$

$\square$

We now show the cumulative regret of the problem. Let

$$\mathbb{I}[\mathbf{y}_X; \mathbf{f}_X] = \frac{1}{2} \sum_{x_i' \in X} \log \left( 1 + \frac{\sigma_{i-1}^2(x_i')}{s^2/\hat{q}_i} \right), \qquad (37)$$

and

$$\gamma_i = \max_{X = [x_1', \dots, x_i'] \subset \mathcal{X}} \mathbb{I}[\mathbf{y}_X; \mathbf{f}_X] \qquad (38)$$

be the maximum possible information gain across $i$ rounds. We then prove the following result.

**Theorem F.5.** *Let $x^* = \arg\max_{x \in \mathcal{X}} f(x)$. With probability at least $1 - \delta$,*

$$\sum_{i=1}^{N} f(x^*) - f(x_i) \le s \beta_N \sqrt{\frac{8N\gamma_N}{q_{\min}}}. \qquad (39)$$

*Proof.* This result is similar to previous results for UCB-based methods, e.g., Theorem 3 in Srinivas et al. (2012) or Corollary 9 in Kirschner & Krause (2018).

With probability at least $1 - \delta$, Lemma F.3 holds. We see that

$$\sum_{i=1}^{N} \left( f(x^*) - f(x_i) \right) = \sum_{i=1}^{N} \underbrace{\left( f(x^*) - \mu_{i-1}(x_i) \right)}_{Corollary\ F.4} + \underbrace{\left( \mu_{i-1}(x_i) - f(x_i) \right)}_{Lemma\ F.3} \qquad (40)$$

$$\le \sum_{i=1}^{N} 2\beta_i \sigma_{i-1}(x_i). \qquad (41)$$

Since

$$\sum_{i=1}^{N} \sigma_{i-1}^2(x_i) \le \sum_{i=1}^{N} \frac{s^2}{\hat{q}_i} \frac{\sigma_{i-1}^2(x_i)}{s^2/\hat{q}_i} \qquad (42)$$

$$\le \frac{s^2}{q_{\min}} \sum_{i=1}^{N} \log \left( 1 + \frac{\sigma_{i-1}^2(x_i)}{s^2/\hat{q}_i} \right) \qquad (43)$$

$$\le \frac{2s^2}{q_{\min}} \gamma_N, \qquad (44)$$

we can rewrite Eq. (41) as

$$\sum_{i=1}^{N} \left( f(x^*) - f(x_i) \right) \le \sqrt{\sum_{i=1}^{N} 4\beta_i^2} \sqrt{\sum_{i=1}^{N} \sigma_{i-1}^2(x_i)} \qquad \text{by Cauchy-Schwartz,} \qquad (45)$$

$$\le \beta_N \sqrt{4N} \sqrt{\sum_{i=1}^{N} \sigma_{i-1}^2(x_i)} \qquad \text{since } \beta_i \le \beta_N, \qquad (46)$$

$$\le s \beta_N \sqrt{\frac{8N\gamma_N}{q_{\min}}} \qquad \text{by Eq. (44).} \qquad (47)$$

$\square$

*Remark* F.6 (The regret bound in Thm. F.5 is lower when $q_{\min}$ increases). We investigate the upper bound in Eq. (47) further to see its dependence on $q_{\min}$.

For simplicity, we will let $\mathbf{K} = k(\mathbf{X}_N, \mathbf{X}_N)$. First, we see that

$$\log \det \left(\mathbf{K} + s^2 \mathbf{Q}_N^{-1}\right) = \log \det s^2 \mathbf{Q}_N^{-1} + \log \det \left(I + s^{-2} \mathbf{Q}_N \mathbf{K}\right) \tag{48}$$

$$\leq \log \det s^2 \mathbf{Q}_N^{-1} + \log \det \left(I + s^{-2} q_{\max} \mathbf{K}\right), \tag{49}$$

which means that

$$\beta_N = B + \sqrt{2 \log \frac{\det \left(\mathbf{K} + s^2 \mathbf{Q}_N^{-1}\right)^{1/2}}{\delta \det \left(s^2 \mathbf{Q}_N^{-1}\right)^{1/2}}} \tag{50}$$

$$= B + \sqrt{2 \log \frac{1}{\delta} + \log \det \left(\mathbf{K} + s^2 \mathbf{Q}_N^{-1}\right) - \log \det s^2 \mathbf{Q}_N^{-1}} \tag{51}$$

$$\leq B + \sqrt{2 \log \frac{1}{\delta} + \log \det \left(I + s^{-2} q_{\max} \mathbf{K}\right)}. \tag{52}$$

Furthermore, we see that

$$\gamma_N = \max_{X=[x_1', \ldots, x_N'] \subset \mathcal{X}} \frac{1}{2} \sum_{x_i' \in X} \log \left(1 + \frac{\sigma_{i-1}^2(x_i')}{s^2 / \hat{q}_i}\right) \tag{53}$$

$$\leq \max_{X=[x_1', \ldots, x_N'] \subset \mathcal{X}} \frac{1}{2} \sum_{x_i' \in X} \log \left(1 + \frac{\sigma_{i-1}^2(x_i')}{s^2 / q_{\max}}\right). \tag{54}$$

Therefore we see that both $\beta_N$ and $\gamma_N$ are upper bounded by terms which are independent of $q_{\min}$, and so the constants in the upper bound provided in Thm. F.5 do not hide any additional dependencies with respect to $q_{\min}$. This shows that the upper bound of cumulative regret from Eq. (47) decays at a rate of $1/\sqrt{q_{\min}}$.

We will now prove the second half of Thm. 4.1. We first prove the following result.

**Lemma F.7.** *Define* $c_1 \triangleq \sqrt{2 \log(2N q_{\max}/\delta)}$. *With probability at least* $1 - \delta$, *for all* $i = 1, \ldots, N$ *and all* $q = 1, \ldots, q_{\max}$, *we have*

$$\left|\bar{y}_{i,q} - f(x_i)\right| \leq \frac{c_1 s}{\sqrt{q}}. \tag{55}$$

*Proof.* From Corollary F.2, we know that the $\bar{y}_{i,q}$ is normally distributed with mean $f(x_i)$ and standard deviation $s/\sqrt{q}$. By Chernoff bounds, for each $i = 1, \ldots, N$ and each $q = 1, \ldots, q_{\max}$, we would have $\bar{y}_{i,q} - f(x_i) > c_1 s/\sqrt{q}$, and $f(x_i) - y_{i,q} > c_1 s/\sqrt{q}$ where either event happens with probability no greater than $\delta N/2q_{\max}$. This means that with probability no greater than $\delta/N q_{\max}$, we have $|y_{i,q} - f(x_i)| > c_1 s/\sqrt{q}$. Therefore, by union bound, we have for all $i = 1, \ldots, N$ and each $q = 1, \ldots, q_{\max}$, we would have $|\bar{y}_{i,q} - f(x_i)| \leq c_1 s/\sqrt{q}$ with probability greater than $1 - (\delta/N q_{\max})(N q_{\max}) = 1 - \delta$. $\square$

**Theorem F.8.** *Define*

$$\tau_q = c_1 s \left(\frac{1}{\sqrt{q_{\min}}} + \frac{1}{\sqrt{q}}\right). \tag{56}$$

*Then, with probability at least* $1 - \delta$, *for all* $i \leq N$, *if we have* $f(x_i) < \max_{j<i} f(x_j)$, *then* $\hat{q}_i < q_{\max}$.

*Proof.* With probability at least $1 - \delta$, Lemma F.7 applies.

To prove the statement above, we show its contrapositive. Suppose we have $\hat{q}_i = q_{\max}$, or that the trial for $x_i$ does not terminate early. This implies that $\bar{y}_{i,q} \geq \max_{j<i} \bar{y}_{j,\hat{q}_j} + \tau_q$ for all $q = q_{\min}, \ldots, q_{\max}$.

For any $q$ in this range, we would then have

$$f(x_i) \geq \bar{y}_{i,q} - \frac{c_1 s}{\sqrt{q}} \tag{57}$$

$$\geq \max_{j<i} \bar{y}_{j,\hat{q}_j} + \tau_q - \frac{c_1 s}{\sqrt{q}} \tag{58}$$

$$\geq \max_{j<i} f(x_j) - \frac{c_1 s}{\sqrt{q_{\min}}} + \tau_q - \frac{c_1 s}{\sqrt{q}} \tag{59}$$

$$= \max_{j<i} f(x_j). \tag{60}$$

This proves the contrapositive which in turn proves the original statement. □

Finally, Thms. F.5 and F.8 can be combined with appropriate union bounds to achieve a more formal version of Thm. 4.1.

### F.3.3 ABLATION STUDIES ON TOY EXAMPLES

In Fig. 14, we provide a brief empirical demonstration of efficiency gains due to early termination. We see that when the noise variance is too high, querying the function once per input would give observations which are too noisy to give good information. Meanwhile, by repeating each query a maximum number of times, we can obtain a good estimate of the true function and allow the BO process to arrive at the optimal using few queries. However, we see that when early termination is allowed, we can still arrive at the optimal input as before, while not requiring all queries to be repeated the maximum number of times. This shows that early termination allows for efficiency gains while minimally sacrificing on the actual optimization process.

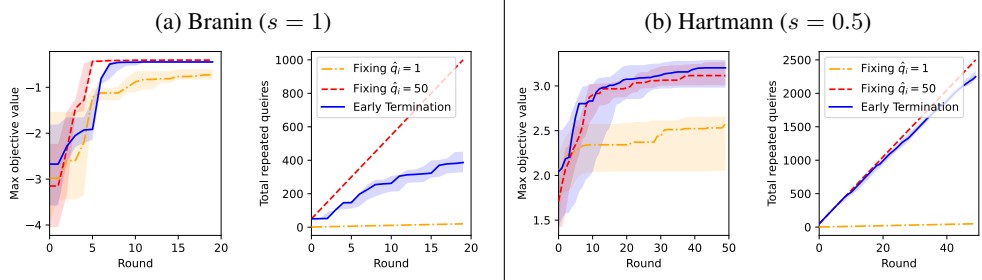

Figure 14: Results of BO with early termination (with $q_{\min} = 1$) on different synthetic functions where each query has different noise levels. For each case, we show the best queried objective value (left plot), and the cumulative number of total repeated queries made with each BO round (right plot).

In practice, since it is difficult to determine $\gamma_N$, $\beta_N$, $s$, and $c_1$ exactly, we instead fix $\beta_i$ and $\tau_q$ to some constant. In OPPA, we choose $\beta_i = 1$ and $\tau_q = 10^{-3}$. We find that these values work well for our methods. Furthermore, in our proofs we do not consider the constrained BO setting. Despite this, the early termination can still be used in practice to achieve good results, which we show in the experiments.

## G  PSEUDOCODE OF OPPA

We present the pseudocode for OPPA in Algorithm 3.

In practice, we find that fixed hyperparameter values (i.e., setting $\beta_t = 1$, $\tau_q = 10^{-3}$, and $q_{\min} = 5$ in all iterations as done in our main experiments) gives consistent results, since the parameters are generally robust to different choices of hyperparameter values anyway.

In the experiments, the value of $q_{\max}$ are set in accordance to Table 2 depending on the benchmark; however in practice can be set as large as the practitioner wants depending on how accurately they would like to measure the actual throughput of the PC. As plotted in Fig. 15, we find that after

a certain point, a larger number of repeats will typically not have large effects on the measured throughput of a PC, showing that OPPA is quite robust to different choices of $q_{\max}$. For OPPA, larger $q_{\max}$ is more tolerable since the early termination mechanism will not exhaust such budget anyway.

We also consider removing outliers which are beyond two times the IQR from the mean; this is be more lenient towards fluctuating training effects, however can also be set to 1.5 times the IQR as typically done as well with little consequences, as described in App. F.2.

---

**Algorithm 3** OPTIMIZER FOR PARALLELISM CONFIGURATIONS (OPPA)

---

1: Generate all valid PCs $\mathcal{H}$
2: **for** $i = 1, 2, \ldots, N$ **do**
3:     **if** $i < N_{\text{random}}$ **then**
4:         Select $H_i$ randomly
5:     **else**
6:         // Step ① – Modeling throughput and memory usage
7:         Construct $\mu_{\mathcal{R},i-1}$ and $\sigma^2_{\mathcal{R},i-1}$ according to Eqs. (24) and (25) respectively
8:         Construct $\mu_{\mathcal{M},i-1}$ and $\sigma^2_{\mathcal{M},i-1}$ according to Eqs. (26) and (27) respectively
9:         // Step ② – Selecting the next PC to query
10:        $H_i \leftarrow \underset{H \in \mathcal{H} \backslash \{H_1, \ldots, H_{i-1}\}}{\arg\max}$ cUCB$_i(H)$ where cUCB$_{i-1}$ is defined in Eq. (4)
11:     **end if**
12:     // Step ③ – Querying some PC
13:     **for** $q = 1, \ldots, q_{\max}$ **do**
14:         Measure training step time as $t_{i,j}$
15:         Compute $\bar{r}_{i,q}$ and $\sigma^2_{\bar{r}_{i,q}}$ according to Eq. (28)
16:         **if** $q \geq q_{\min}$ and $\bar{r}_{i,q} < \max_{j<i} \bar{r}_{j,\hat{q}_j} + \tau_q$ **then**
17:             **break**
18:         **end if**
19:     **end for**
20:     $\hat{q}_i \leftarrow q$ to track the number of training steps ran in round $i$
21:     Measure maximum memory usage as $m_i$
22:     **if** time budget exceeded **then**
23:         **break**
24:     **end if**
25: **end for**
26: **return** $H_{i^*}$ where $i^* = \arg\max_{i \leq N:(m_i < M_0) \wedge (\hat{q}_i = q_{\max})} \bar{r}_{i,\hat{q}_i}$

---

# H ADDITIONAL INFORMATION ON EXPERIMENTAL SETUP

## H.1 TRAINING AND HARDWARE CONFIGURATIONS

We list the models used in our experiments in Table 2, along with the allotted search time and how many trials we repeat on them. All models used are based on the transformer architecture, and were retrieved from Huggingface.

Note that in all of our plots, we plot the median value (with a line) and also the lower and upper quartiles (with a fainter band over and under the line). We do so since we find that the values are often asymmetrically skewed, and therefore opted to show the quartile values to more accurately represent the distribution of these values. Also note that the repeated trials were reduced for larger models due to restrictions in compute budget.

Table 2: Details of models used in our experiments and the corresponding training scenario

| Case | Model | # Params | Batch size | Max. seq. length | Search Time | $q_{\max}$ | Repeats |
|------|-------|----------|------------|------------------|-------------|-----------|---------|
| BERT | BERT Base Uncased | 110M | 256 | 256 | 20 mins | 50 | 10 |
| Qwen | Qwen-2 | 1.5B | 64 | 1024 | 20 mins | 30 | 10 |
| LLaMa-1b | LLaMa 3 | 1B | 64 | 1024 | 60 mins | 20 | 5 |
| LLaMa-7b | LLaMa 2 | 7B | 256 | 1024 | 60 mins | 30 | 5 |

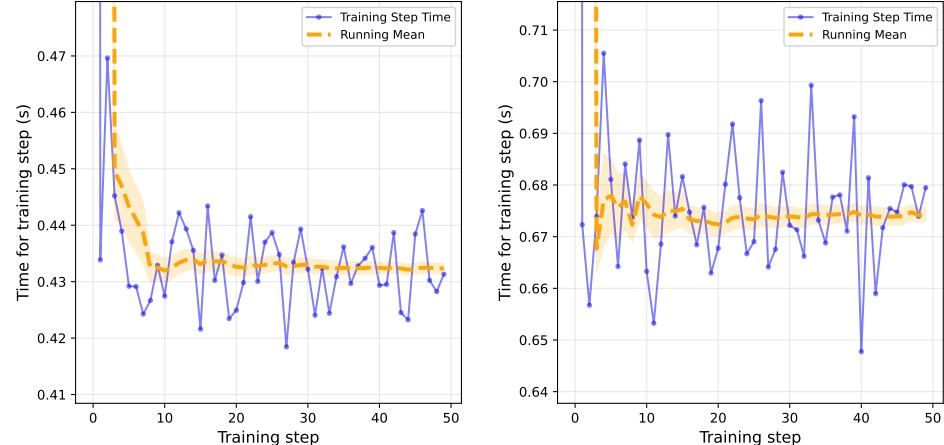

Figure 15: The effect of number of training steps repeatedly measured on the predicted throughput. The blue lines represent a training trial and the measured time per training step. The orange line represents the average training time (i.e., reciprocal of throughput) predicted up to that certain training step.

Our model training is implemented based on the COLOSSAL-AI framework (Li et al., 2023), which allows execution of NN training with 3D parallelism with different tunable hyperparameters. We note, however, that OPPA is also general enough to be applied to any other training framework as well, whose implementation we leave to future works.

In Table 3, we list the hardware configuration used in our experiments. The hardware configurations used are based on the resources that are available to the authors.

Table 3: Configurations of tested hardwares.

| Config. Name | GPU Model (Memory) | GPUs per host | # Host | Multi-Host Characteristic |
|---|---|---|---|---|
| 8 GPUs | NVIDIA RTX A5000 (24GB) | 8 | 1 | |
| 16 GPUs | NVIDIA RTX 3080 (10GB) | 8 | 2 | Docker Overlay Network |
| 32 GPUs | NVIDIA A100 (40GB) | 4 | 8 | High-Performance Compute |

## H.2 ALGORITHMS RAN

We list the algorithms we have ran along with their implementation details here.

- RANDOM. This involves randomly selecting a PC from $\mathcal{H}$ to trial in each round until the time budget is exhausted.

- XGBOOST (Chen & Guestrin, 2016). This is the method DEEPSPEED (Rasley et al., 2020) uses to configure the PC, and is adapted to work with the hyperparameters in our PC. The method involves training an XGBOOST model based on the observed throughput values, then selecting the next PC to trial as the one whose predicted throughput is the highest. This is repeated until time budget is exhausted.

- COST-MODEL. This involves using the cost model as described in Apps. D.1 and D.2, learned based on several randomly selected PCs, to obtain an estimate of throughput and maximum memory usage, and perform a one-shot selection of the best PC according to the predictions.

- VANILLA-BO. This performs BO whose GP has constant mean and Matern kernel with $\nu = 5/2$. The cUCB criterion is used for PC selection. The BO loop is implemented using BOTORCH (Balandat et al., 2020).

- OPPA. This is the method proposed in Sec. 4, which involves modifying BO to include a parallelism-informed prior belief and early trial termination. The hyperparameters used are as stated in App. G.

We note that due to the search space employed for $\mathcal{H}$, we do not consider benchmarks which performs non-adaptive optimization with a cost model. This is because those methods optimize with respect to the computation graph rather than the hyperparameters which we discussed in App. C.1, and since they do not use the same information as OPPA to perform optimization, making it futile to compare between the two since they focus on optimizing different aspects of training parallelism. Nonetheless, we provide some comparisons with selected algorithms of such nature, namely Li et al. (2022) and Lin et al. (2024) in Fig. 8.

# I ADDITIONAL RESULTS

## I.1 PLOTS OF BEST ACHIEVED THROUGHPUT VERSUS OTHER QUANTITIES

In Fig. 16, we plot the best achieved throughput versus the number of training trials that have been ran. We see that in this view, OPPA still outperforms other benchmarks. While the margin may be smaller in some instances, we see that OPPA is able to achieve the good results more consistently as seen by the error bars when compared to some of the other methods. This also demonstrates that the prior belief used in OPPA alone would have helped in achieving a better performance regardless of the early termination mechanism in OPPA.

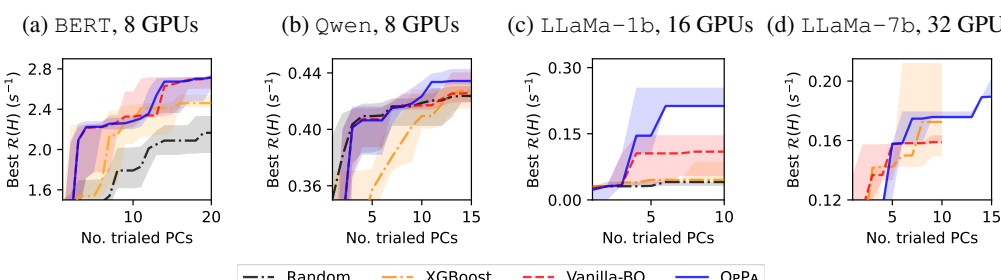

Figure 16: Results of the best obtained throughput (higher is better) plotted against the number of PCs that have been trialed. The lines represent the median value of the best obtained throughput across five trials, while the error bar represent the quartile values. Note that for the experiments on 32 GPUs, we are unable to run the optimization beyond the allotted time due to resource constraints and therefore are only able to plot some algorithms for a fewer number of queried PCs.

In Fig. 17, we show the achieved throughput is plotted against the number of training steps performed in the training trials, showing that the difference in efficiency of OPPA becomes more pronounced. From these results see that OPPA is both more time efficient and query efficient, which can be useful when the overhead to perform one trial may become higher, for example when the framework is adapted to run on a cluster with a job scheduler.

## I.2 OPTIMAL PCS RECOVERED BY OPPA

In Table 4, we show the PCs that were recovered by OPPA. Note that the optimal PCs chosen match quite well with intuition, where for smaller models DP tends to be prioritized. Meanwhile for larger models and training scenarios which are done across multiple machines, PP is prioritized.

## I.3 PC OPTIMIZATION FOR OTHER MODELS

In Fig. 18, we presented the results for tuning the PC for ViT model (Dosovitskiy et al., 2021). We see that the results here show that OPPA is able to select better PCs compared to the other methods, consistent with other results in the paper.

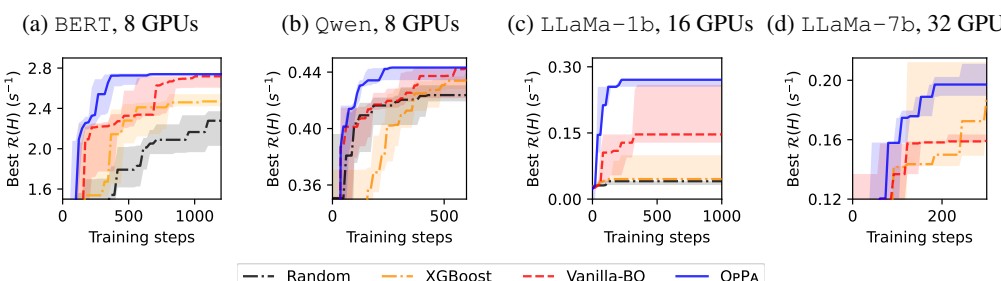

Figure 17: Results of the best obtained throughput (higher is better) plotted against the total number of training steps ran during in the training trials. The lines represent the median value of the best obtained throughput across five trials, while the error bar represent the quartile values.

Table 4: Example of optimal PCs selected by OPPA in different training scenarios. Values are based on observations of optimal PCs across multiple repeated trials. A range of value shows that a certain parameter shows a spread across multiple trials (i.e., no strong preference towards one value), while a dash shows that the parallelism dimension associated with that hyperparameter is not used.

| Hyperparameter | BERT, 8 GPUs | Qwen, 8 GPUs | LLaMa-1b, 16 GPUs | LLaMa-7b, 32 GPUs |
|---|---|---|---|---|
| DP size (dp) | 8 | 4 | 1 | 2 |
| TP size (tp) | 1 | 1 | 1 | 1 |
| PP size (pp) | 1 | 2 | 16 | 16 |
| DP bucket size | $1-64$ | $1-4096$ | - | $1-4096$ |
| ZeRO stage | 0 | 1 | - | 1 |
| ZeRO bucket size | - | $1-64$ | - | $64-4096$ |
| Overlap ZeRO communication | - | True/False | - | False |
| Overlap ZeRO AllGather | - | True/False | - | False |
| # microbatches (mb) | - | 8 | 64 | 32 |
| # model chunks (mc) | - | $1-2$ | 1 | 1 |
| Overlap P2P for PP | - | True/False | False | True/False |
| Grad. checkpointing | True | False | False | False |

Additionally, in Fig. 19, we presented the results for tuning the PC for transformer-based Mixture-of-Experts (MoE) model Shazeer et al. (2017), with the additional tuning of the expert parallelism (EP) degree being performed by the same framework in addition to the existing DP, TP and PP. For the experiments we consider parallel training of a model with 16 transformer units, 32 attention heads and 32 experts. Each trial is ran for up to $q_{max} = 50$ steps, and in OPPA, we set $q_{min} = 10$ to account for higher variation in training step times. In the results, we again see consistently better performances compared to the other competing methods. Specifically, even if OPPA may eventually lead to similar results to vanilla BO at the end of the optimization process, OPPA is able to do so more consistently (lower variance across random runs), and is able to arrive at the optimal PC much more rapidly and efficiently. These results further demonstrate that OPPA is able to also generalize to other models as well.

### I.4   PREDICTED THROUGHPUT AND MEMORY BY PARALLELISM-INFORMED PRIOR BELIEF

In Figs. 20 to 22, we compare the modeled throughput with the true values in different training scenarios. We see that in this case, using the prior belief allows for the values to be modeled adequately well, but more importantly, allow for the PC which achieves the best throughput to also have the highest values, and therefore be identified correctly. We find that for the BO process, a surrogate only needs to model the good PCs well in order to select a good PC in the end. Meanwhile, the GP without prior belief learns the patterns much less efficiently or do not learn them at all. This correlates well with the results in the main text where standard BO selects a worse PC compared to OPPA which uses a better prior belief.

In Fig. 23, we compare the modeled maximum memory with the true measured value. In both cases a GP has been used however with and without a parallelism-informed prior belief since only VANILLA-BO and OPPA are the only benchmarks we tested which explicitly models the memory

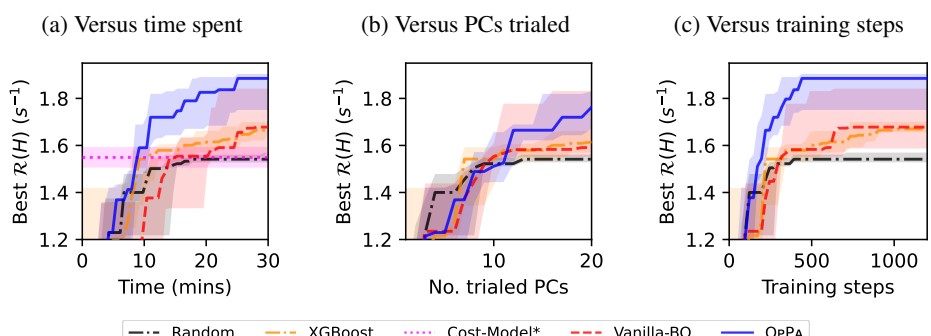

Figure 18: Results for training ViT model (Dosovitskiy et al., 2021) with batch size of 256.

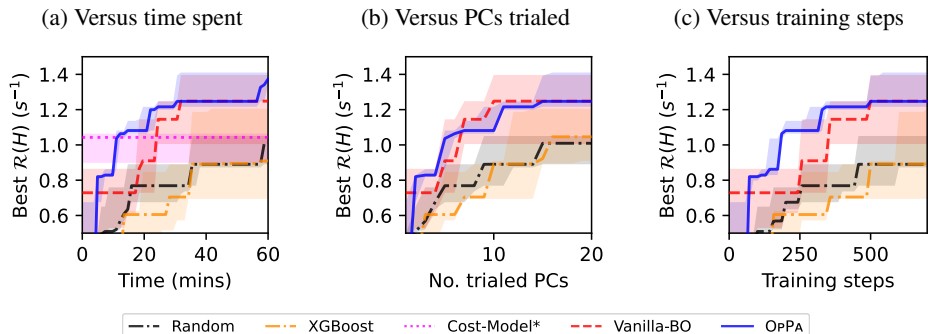

Figure 19: Results for training Mixture-of-Expert (Shazeer et al., 2017) models. Note that we do not run all benchmarks due to resource constraints.

usage. Here, we see that OPPA is able to better model the memory usage due to its use of the prior belief. This is reflected in the confusion matrices which shows that after training, OPPA is able to more accurately detect when a certain PC will result in out-of-memory errors.

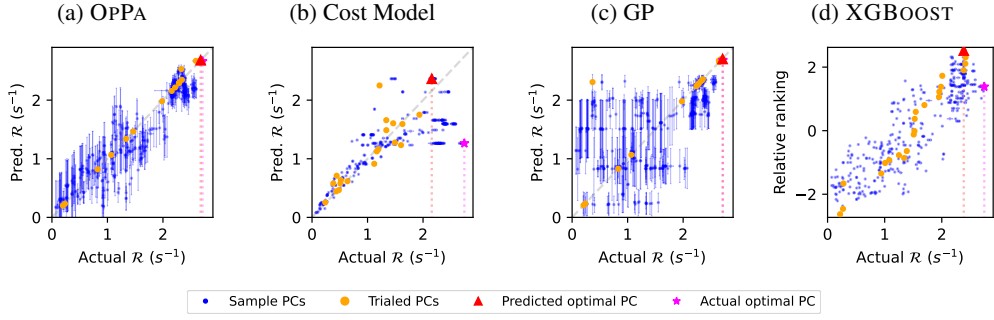

Figure 20: Comparison of modeled throughput values versus the true throughput for training of `BERT` model on 8 GPUs after 20 PC trials.

To additionally demonstrate the interpretability of the GP modeling for the surrogate, in Table 5, we show the results for the lengthscales learned by the kernel (as given in Eq. (22)). We see that hyperparameters that have larger effects on the resulting throughput or are less well-modeled by our prior belief will typically correspond to the shorter lengthscales. For example, for the training of BERT, we see that the parameters for TP dimension size and for the number of microbatches (for PP) have shorter lengthscales. This matches our intuition where the throughput would be more sensitive to the increased TP or PP being used (for the worse). This additional interpretability makes GP a

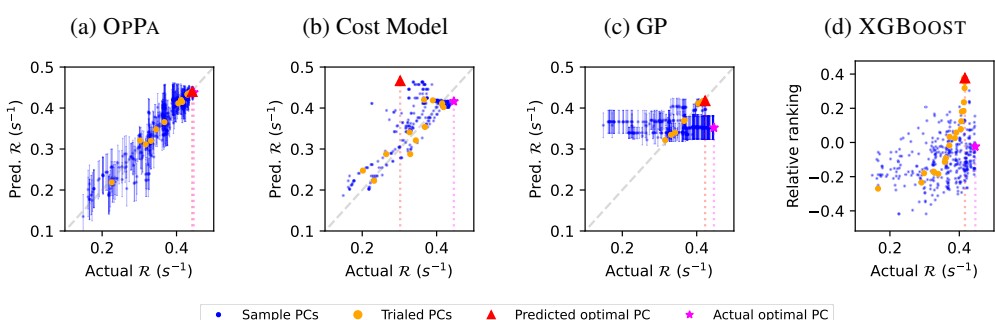

Figure 21: Comparison of modeled throughput values versus the true throughput for training of `Qwen` model on 8 GPUs after 20 PC trials.

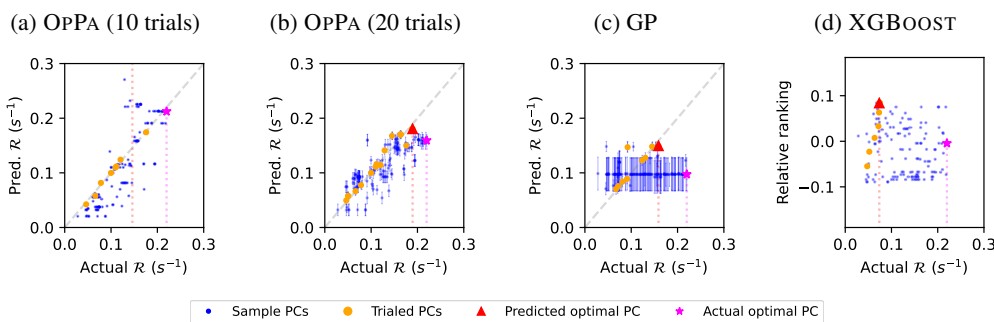

Figure 22: Comparison of modeled throughput values versus the true throughput for training of `LLaMa-2` model on 32 GPUs for, in order, OPPA after 10 trials, OPPA after 20 trials, GP after 10 trials and XGBOOST after 10 trials. Note that among the three algorithms only OPPA ran for up to 20 trials given the time constraint.

very suitable candidate for surrogate modeling in this case, since it allows practitioners to be more aware of the modeling intuition by the surrogate as well as being accurate.

Table 5: Example of the log lengthscales learned by the kernel of the GP for the throughput surrogate model. The bolded value are to highlight hyperparameters with particularly shorter lengthscales.

| Hyperparameter | `BERT`, 8 GPUs | `Qwen`, 8 GPUs |
|---|---|---|
| DP size (`dp`) | 1.877 | 5.233 |
| TP size (`tp`) | **-1.071** | 6.024 |
| PP size (`pp`) | 2.293 | **-1.849** |
| DP bucket size | 3.484 | 5.994 |
| ZeRO stage | 0.434 | 5.279 |
| ZeRO bucket size | 3.402 | 6.516 |
| Overlap ZeRO communication | 3.949 | 7.581 |
| Overlap ZeRO AllGather | 3.822 | 7.589 |
| # microbatches (`mb`) | **-1.322** | **-1.741** |
| # model chunks (`mc`) | -0.293 | 0.124 |
| Overlap P2P for PP | 3.854 | 6.974 |
| Grad. checkpointing | -0.334 | 6.586 |

## I.5    EFFECTS OF PRIOR MISSPECIFICATION

To investigate the robustness of our method with respect to a misspecified prior, we conduct experiments to see how OPPA behaves as our cost-model prior becomes increasingly inaccurate. To do so, we add a perturbation term into our cost function, where we increase the magnitude of the perturbation term up to about 25% and 50% that of the maximum throughput obtained. We present the median obtained throughput across 5 random trials in Table 6. We see that even when the cost-function prior

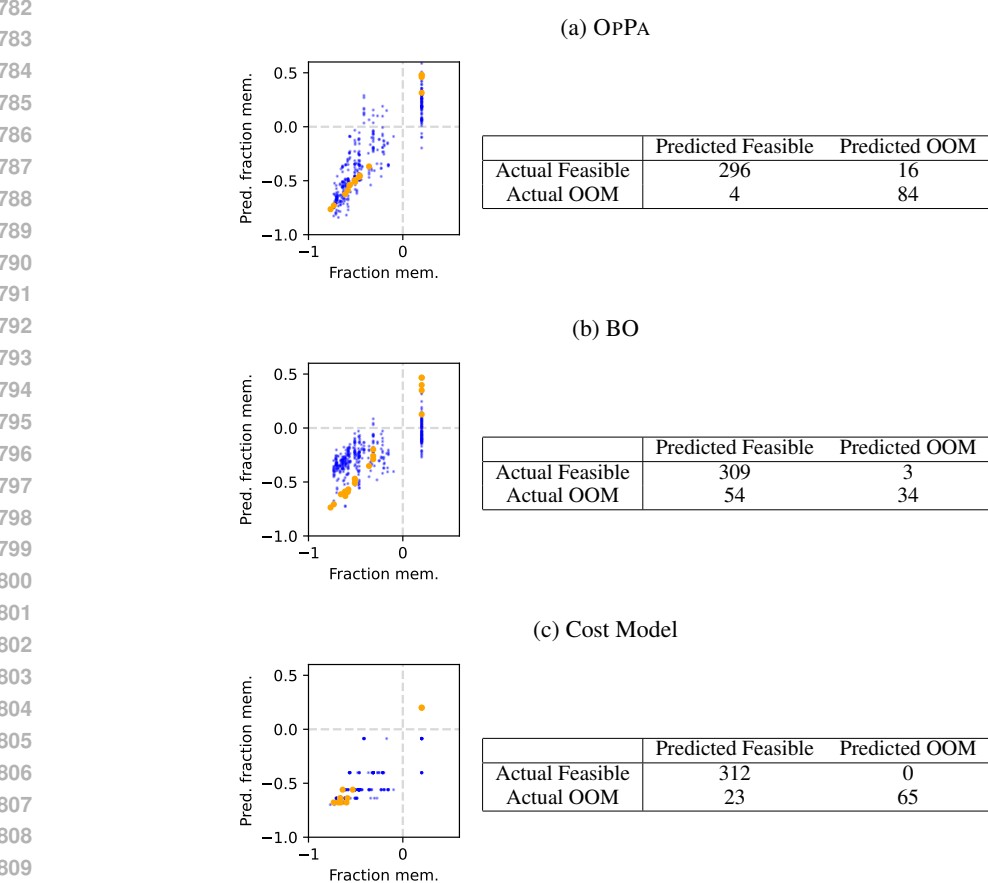

(a) OPPA

| | Predicted Feasible | Predicted OOM |
|---|---|---|
| Actual Feasible | 296 | 16 |
| Actual OOM | 4 | 84 |

(b) BO

| | Predicted Feasible | Predicted OOM |
|---|---|---|
| Actual Feasible | 309 | 3 |
| Actual OOM | 54 | 34 |

(c) Cost Model

| | Predicted Feasible | Predicted OOM |
|---|---|---|
| Actual Feasible | 312 | 0 |
| Actual OOM | 23 | 65 |

Figure 23: Comparison of modeled maximum memory usage versus the true maximum memory usage for training of `Qwen` model on 8 GPUs after 20 PC trials. In each row, the left hand graph shows the predicted value versus the actual value (with the predictive variance are omitted for clarity), while the right hand table is the confusion matrix.

is adversarially constructed (by knowingly adding an incorrect term into the cost function), we are still able to obtain good performances to the unperturbed cost-model prior even if the convergence is slightly slower. This suggests that even in this extreme case, the GP is able to correct for the inaccuracies in the cost-prior effectively.

In practice, prior misspecification typically will due to the cost function not being sufficiently complex to match the true parallel training dynamics, because of incomplete or inaccurate domain knowledge about the true system rather than due to an adversarial construction of the cost function. This is the case in the cost function we have chosen in our paper, where there is a discrepancy between the cost function alone and the true throughput as demonstrated in Fig. 6c, resulting from our cost function not modeling the effects of all hyperparameters in the PC. Under practical scenarios, we therefore would not expect the results to be as extreme as what we have seen in the presented results, and that a GP should be able to effectively model the throughput values.

Table 6: Effect of perturbing the prior belief on the resulting optimization process. The values reported are the median throughput obtained for the PC found after certain number of minutes of running OPPA with the perturbed prior belief function.

| Percent perturbation magnitude | 10 mins. of search | 20 mins. of search |
|---|---|---|
| 0 (original prior) | 0.425 | 0.446 |
| About 25 | 0.424 | 0.447 |
| About 50 | 0.415 | 0.447 |

## I.6 ADDITIONAL RESULTS FOR ABLATION STUDIES OF COMPONENTS IN OPPA

In Fig. 24, we demonstrate how early termination and parallelism-informed prior belief affect the overall achieved throughput. First, we see that when parallelism-informed prior belief is used, the performance is no worse than when no prior belief is used, although this benefit is more pronounced for the Qwen example, possibly due to the increased complexity in the training setup. meanwhile, with early termination, we see that the performance is better in terms of time and number of training steps needed, while not sacrificing the performances when considering the number of PCs that are trialed to achieve a certain performance. This shows that the benefit gain comes from being able to shorten the duration of the training trials while not sacrificing the throughput predictions.

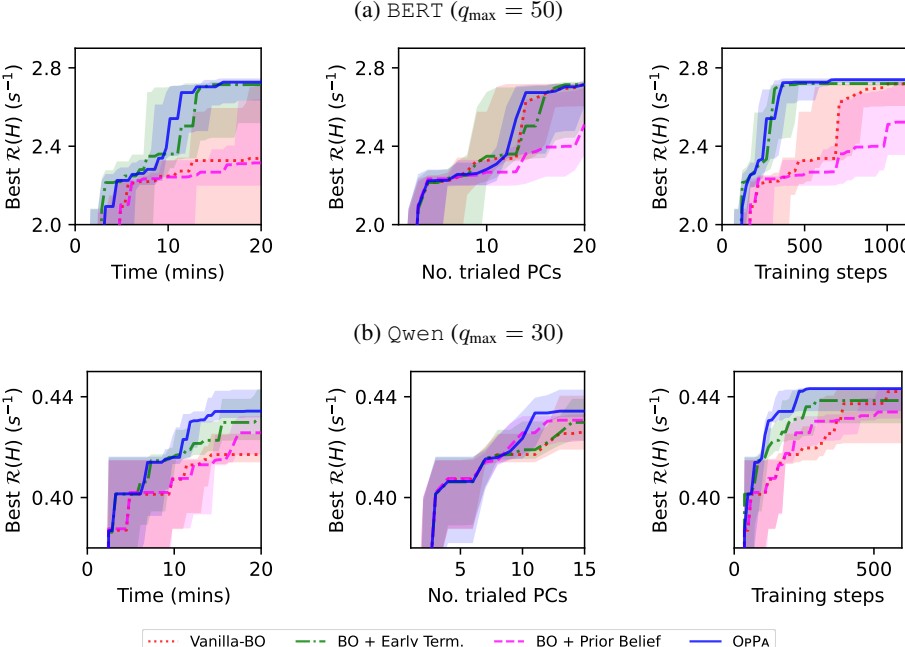

Figure 24: The effects of different components of OPPA on the optimal PC found. In each row, we present the results in a certain training setup, where we present the obtained throughput versus, from left to right, the time the algorithm has been ran for, the number of unique PCs trialed, and the number of training steps that has been ran across all of the trials.

## I.7 THE EFFECT OF $q_{min}$ ON OPPA PERFORMANCE

In Fig. 25, we see how the choice of $q_{min}$ affects the achieved throughput. For the BERT case, we see that early termination clearly improves the time required for optimization, as seen where when $q_{min} = q_{max}$ the obtained PC is the worst when all methods are allotted the same amount of time. As $q_{min}$ decreases, we see that there is less drop in the amount of time required since each training step is dominated by the time to setup each PC trial. However, we still see that when we plot the number of training steps for the optimization, we see that a smaller $q_{min}$ will require fewer training steps to arrive at the same optimal PC. However, this trend breaks down when $q_{min}$ is too small, likely since the value obtained is too noisy to give good information. Nonetheless, even in this case, we stil obtain good results. For the Qwen case, similar trends can be seen where reducing $q_{min}$ is able to reduce the time and the number of training steps required to find the optimal PC up to a certain point.

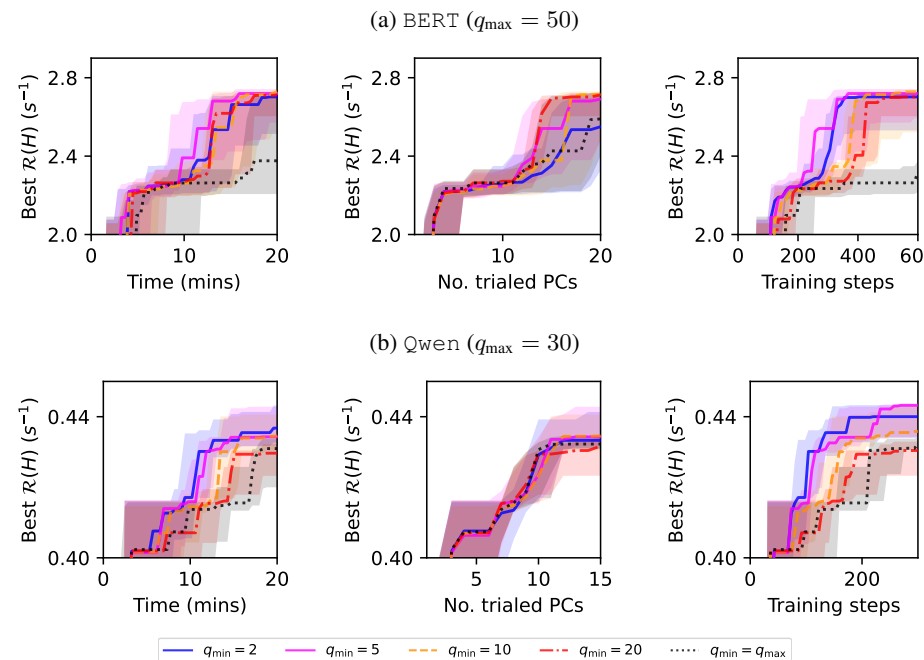

Figure 25: The effects of $q_{min}$ on the optimal PC found. In each row, we present the results in a certain training setup, where we present the obtained throughput versus, from left to right, the time the algorithm has been ran for, the number of unique PCs trialed, and the number of training steps that has been ran across all of the trials.

# J  ADDITIONAL DISCUSSIONS REGARDING THE PAPER

## J.1  REGARDING THE NOVELTY OF THE PROPOSED METHOD

Here, we highlight some of the novelty of our work beyond being a direct application of Bayesian optimization (BO), which sets it apart from these works. This has been highlighted in Secs. 3 and 4, and is done so by (1) using specific characteristics within the PC optimization problem to inform different design choices for the BO process, and (2) developing a novel BO technique with provable theoretical guarantees, which is an advancement for BO in itself.

First, we point out that *our work identifies specific characteristics of the PC optimization which allows us to design* OPPA *to directly tackle these points and obtain strong performances*. Unlike a typical hyperparameter optimization problem, we notice that there are several ways in which optimizing the PC differs, each allowing us to incorporate appropriate and novel techniques into our framework beyond using vanilla BO with an updated prior function.

- **Designing an appropriate surrogate model and prior for PC optimization which works across many training scenarios is non-trivial** (Sec. 4.1). In our problem setting, we designed a prior function which is specific enough to capture characteristics of parallelized NN training based on existing domain knowledge, while still allowing enough flexibility to adapt to a wide range of possible parallel training scenarios. This is demonstrated in our results where the same choice of model and prior is valid on a variety of hardware configurations, and give strong results across all of them.

- We also utilize **black-box constraints to filter out infeasible PCs from the search space** (Secs. 3 and 4.2). In PC optimization, the black-box constraint will naturally arise since some unknown PCs may result in OOM errors on real machines. We therefore attempt to automatically learn the feasible space and inform the search process accordingly. This is unlike vanilla BO which would assume that the feasible set is fully known, and perform search accordingly.

- Since trialing a PC involves sequentially running many training steps to measure the running time, we also **introduce a novel and systematic way to detect suboptimal PCs early and terminate them to save time** (Sec. 4.3). This is designed based on how training throughput is measured, and whose novel solution is proposed and analyzed (both theoretically and empirically), as we discuss in the next point.

Second, we *introduce a novel BO algorithm* where an experiment is sequentially repeated, and can be terminated early in suboptimal cases (Sec. 4.3 and App. F.3). We *inspire this problem via the optimization of PC in parallel training*, and *introduce a principled method to decide when early trial termination should be done*, providing both theoretical justification and empirical results in PC optimization. This improves on existing BO works which will fix the number of times an experiment should be repeated before the experiments are performed, and will waste resources on suboptimal trials. Outside of PC optimization, our method can also be adapted to cases where noisy measurements should be recorded multiple times to obtain a better estimate such as in real scientific experiments where trials are often repeated anyway.

### J.2 REGARDING THE NECESSITY OF REAL TRAINING TRIALS

We note that while the effect of some hyperparameters in a PC can be estimated reasonably well, this would not be the case for all hyperparameters (as already stated in Secs. 2 and 3). The only method to accurately tune these hyperparameters would therefore be based on empirical data from actual model training. To achieve the best performance for NN training, it is therefore inevitable that we would have to perform some actual NN training to evaluate the effects of these hyperparameters. This motivates why adaptive methods which rely on actual time for training is important. This viewpoint is reflected in real parallel training frameworks such as DEEPSEED or NEMO where real training trials are also used to perform PC tuning (as also highlighted in Sec. 2), and also further demonstrated throughout in our paper to be superior to non-adaptive methods (e.g., as seen in Fig. 8).

Furthermore, in practice, the training time is often long and a relatively short time for PC optimization can already give large savings on the overall efficiency of NN training. To more concretely demonstrate this, in Fig. 5b, we have presented the benefits of optimizing the PC with OPPA before performing actual training. In our case, the optimization process is done for less than an hour which in practice, is insignificant compared to the time for training large-scale models. For instance, finetuning a language model may take several hours or days, while pretraining from scratch often extends to weeks or even months. When we extrapolate the speed of training process to see how many training steps can be processed in a few hours, as done in Fig. 5b, we find that the PC chosen by OPPA can already lead to many more training steps being performed compared to other methods, or even compared to using a cost model alone to non-adaptively select a PC (which will still require time to perform optimization nonetheless). This shows that in many practical scenarios where the actual training would be done for a long period of time, a relatively short time spent on optimizing the PC can lead to large benefits in computational saving.

## K LIMITATIONS AND BROADER IMPACTS

In this work, we have mainly focused on optimizing tunable hyperparameters which are found in common parallel training frameworks. While there are many other aspects and search spaces of parallelization that we could consider, we have instead mainly considered hyperparameters which would generally be tuned manually by practitioners who want to perform parallel training. We believe that Bayesian optimization with an appropriate formulation could also allow our methods to these other search spaces that may arise as well. Maximizing the throughput during NN training would allow the same amount of computation to possibly be done in a more efficient manner, both in terms of time and compute resources. While this may allow faster development of NNs for both good and bad use cases, overall it would still have a positive impact since it allows for higher efficiency which reduces waste in computation time and other feasible resources that come with it.

