# OpenReview forum: "Configuring Parallel Training of Neural Networks using Bayesian Optimization"
_ICLR.cc/2026/Conference — Submitted to ICLR 2026_

### Official Review · Reviewer_jA8L · 2025-10-30

**Soundness:** 2
**Presentation:** 2
**Contribution:** 2
**Rating:** 4
**Confidence:** 3

**Summary:**

This work introduces a Baysian optimization method to find suitable hardware parallelization configurations using a minimal amount of execution time. Notably, the surrogate Matern Gaussian Process can integrate prior guesses about the throughput and memory use that are refined during the subsequent optimization. The optimizer is experimentally evaluated on Transfomer architectures for multi-GPU settings, demonstrating reduced optimization time compared to simpler baselines.

**Strengths:**

- The work tackles an important practical problem is distributed training with a focus on overall optimization time
- The ability to integrate prior domain knowledge is likely beneficial in this context as it is plausible that good guesses about memory use and execution time are available for a wide variety of hardware and models.
- The manuscript includes an introductary overview of techniques and related prior work, making it accessible and self-contained

**Weaknesses:**

- The novelty of the method is limited as the employed techniques for Bayesian surrogate optimization are well established
- The memory component of the optimization problem is modelled as part of the constraint and is thus only implicitly optimized for. It seems more suitable to formulate the problem as an multi-objective optimization problem to allow the optimizer to explicitly optimize the memory.
- The idea of early termination for likely suboptimal configuration warrants further investigation since a search sample's usefulness is not necessarily determined by its closeness to an 'optimal' value. Since the suboptimal value was suggested by the surrogate model it represents a point in the search space that the surrogate model predicts to be better than it actually is. This means that this suboptimal sample will likely be very useful for improving the surrogate model. Of course, there is a practical trade-off between taking time to generate a sub-optimal sample and taking time to generate many samples but it seems that the early termination throws away information that could be useful to improve the surrogate model (and thus speed up overall convergence). The ablation study in Figure 7 suggests that both Early Termination and Prior Belief are required so it could be that Early Termination is helpful if the Prior Belief already insures that the surrogate model has a good idea of the promising regions in the search space. The presented experiments do not seem to delineate between 'time saved through convergence speed' and 'time saved through termination'.
- The search method itself seems to rely on hyperparameters (e.g. IQR range for outlier detection) but it unclear how they are chosen in practice

**Questions:**

- L237 assumes "All-Reduce operations involved in DP and TP, and point-to-point communications involved in PP. We consider a canonical ring/tree scaling with hierarchical aggregation such that intra- and interhost connections are separately modeled" -- do these assumptions matter for the ability to provide domain knowledge priors or do you expect OpPa to generalize to other settings?
- L294 "we assume the additional contribution is expected to have no additional bias on the estimates of the quantities" - Can you please elaborate on why this is justified.

---

> ### Author Response · Authors · 2025-11-25
> **Response (Part 1/3)**
>
> We thank the reviewer for their comments, particularly with their positive feedback regarding the "practical problem" being tackled and the ability of OPPA to "integrate prior domain knowledge", and how the paper is "accessible and self-contained".
>
> Below, we provide responses to the points raised by the reviewer.
>
> ---
>
> > The novelty of the method is limited as the employed techniques for Bayesian surrogate optimization are well established
>
> We note that while Bayesian optimization (BO) is already well-established, our work is still novel since (1) we have **exploited novel mechanisms which leverage specific aspects of the optimization of parallelism configurations (PCs)** not considered previously, and (2) we have **integrated a novel early termination mechanism into our BO algorithm to boost its efficiency  while preserving  theoretical performance guarantees**, which has not been considered in previous BO works.
>
> Regarding the first point, we claim that OPPA is able to effectively exploit novel mechanisms introduced in Section 4 which leverage specific characteristics of the PC optimization problem, as we have also discussed in the paper (such as in Sections 2 and 3, and in Appendix J.1).
> - First, while the effect of a PC towards the training throughput is somewhat understood, it cannot be modeled perfectly, which suggests that domain knowledge in the form of a prior belief can aid the optimization process. However, designing the appropriate prior belief for the optimization process is not trivial.
>     - As discussed in the Related Works section, **this viewpoint of combining measurements of the training throughput with prior knowledge is not previously studied** within the PC optimization literature. OPPA is then valuable in this aspect as it is the first work to provide a clear framework for how this can be done within this domain.
>     - Furthermore, **it is not obvious what form of the prior belief is suitable in the case of PC optimization**. In OPPA, we presented a prior belief function which is able to sufficiently capture the trends within the parallelized training process, while also being general enough to work across a wide range of NNs and hardware setups. Here, we are able to demonstrate how incorporation of appropriate domain knowledge representation with real measurements can in fact be useful for PC optimization.
> - Second, to measure the throughput, a PC is trialed by running a sequence of training steps and averaging the measurements across these repeated training steps. With these repeated steps, it can become inefficient and we therefore consider whether fewer repeats could have been run while still recovering the optimal PC in the end. Motivated by this viewpoint, early termination mechanism of suboptimal trials could be performed to reduce the number of training steps required while also still recovering the optimum PC.
>
> Regarding the second point, to further improve the efficiency of PC optimization, **OPPA exploits the mechanism for early termination of suboptimal trials to reduce the number of sequential repeated measurements needed to be run**, and within this paper **has been justified theoretically via the guarantees of sublinear regret, and verified empirically in the experiments**. In practice, this mechanism is useful beyond PC optimization, in scenarios such as scientific experiments which may have to be conducted sequentially and repeated in order to obtain more accurate measurements of the true function value.
>
> We appreciate the reviewer for their comments. In response to this, we have included additional clarification in the Introduction and Related Works sections to further highlight the novelty of OPPA from the algorithmic and the practical standpoints.
>
> ---
>
> > The memory component of the optimization problem is modelled as part of the constraint and is thus only implicitly optimized for. It seems more suitable to formulate the problem as an multi-objective optimization problem to allow the optimizer to explicitly optimize the memory.
>
> We thank the reviewer for their suggestion. We clarify that in our formulation, we consider the GPU memory usage as a constraint rather than as an alternate objective **since in typical training scenarios, the user would usually be given the hard constraint on what GPU usage is available**. This would typically be a physical limit from the computation of the GPU, where the user cannot train if the memory usage exceeds that possible by the GPU, otherwise they will encounter an out-of-memory error. In most of these training scenarios, since the GPU memory is given as a hard constraint, we formulate our optimization problem as such as well. However, we do agree that trading off between throughput and required memory can be an interesting angle to explore in future works.
>
> ---
>
> **[Continued Below]**

---

> ### Author Response · Authors · 2025-11-25
> **Response (Part 2/3)**
>
> > The idea of early termination for likely suboptimal configuration warrants further investigation since a search sample's usefulness is not necessarily determined by its closeness to an 'optimal' value. Since the suboptimal value was suggested by the surrogate model it represents a point in the search space that the surrogate model predicts to be better than it actually is. This means that this suboptimal sample will likely be very useful for improving the surrogate model...
>
> We first clarify that in the case when a trial terminates early, **the measured throughput is still used**. However, the variance of the measured throughput will be higher due to fewer repeated trials. Early termination of a trial therefore does not determine whether a certain configuration is considered by the surrogate or not, but instead controls the fidelity of the measurement used in the surrogate. **This is reflected in Line 15 of Algorithm 5 in Appendix G** where all throughput measurements are used regardless of whether early termination occurs, and in Equations 24-27 which do not exclude measurements from early terminated trials in the update equation.
>
> Given this, in order to justify the early termination mechanism, we claim that **after a certain point, repeated querying of a suboptimal configuration will have little effect on the modeled GP surrogate, the belief on the optimal configuration, and the belief on the maximum throughput value**. This observation is demonstrated in the newly-added Appendix F.3.1 and Figures 11-13 in the updated paper. This means that while configurations that are less optimal still carry information about the throughput of different PCs, they will contain less information about the throughput of optimal PCs, and therefore do not need to be trialed for as many training steps while still allowing OPPA to retain sublinear cumulative regret, as stated in Theorem 4.1 and proven in Appendix F.3.2.
>
> To further study the effect of early termination, we perform additional experiments on synthetic functions where early termination is performed: Figure 14 in Appendix F.3.3 shows that the overall performance of the BO process does not degrade much compared to when early termination is not used, while being able to reduce the number of repeated queries that needs to be made overall. **Appendix F.3 therefore demonstrates intuitively, theoretically, and empirically that early termination does not deteriorate the BO process much**.
>
> We appreciate the reviewer for their insightful comment. The discussion on why early termination of suboptimal trials intuitively works has been added to the paper under Appendix F.3.1 as previously mentioned, and we will look to shift them to the main paper when space permits.
>
> ---
>
> > The ablation study in Figure 7 suggests that both Early Termination and Prior Belief are required so it could be that Early Termination is helpful if the Prior Belief already insures that the surrogate model has a good idea of the promising regions in the search space. The presented experiments do not seem to delineate between 'time saved through convergence speed' and 'time saved through termination'.
>
> We note that for the ablation study in Figure 7, **we have performed experiments where only one of the additional adjustments are applied, both individually showing improvements over vanilla BO**. Specifically, we have presented results for the case where only early termination of suboptimal trials is done but the prior belief is not used, _and_ also for the case where prior belief is used but early termination is not done. In both of these cases, we see improvements over vanilla BO. Additionally, **by using both adjustments as in OPPA, we are able to obtain maximum performance**. These results show that the performance gains of OPPA can be attributed to both of these adjustments in the algorithm, rather than to just one of them.
>
> ---
>
> > The search method itself seems to rely on hyperparameters (e.g. IQR range for outlier detection) but it unclear how they are chosen in practice
>
> In our experiments, we have found that **OPPA is robust to the choices of different hyperparameters**. This includes the IQR range mentioned by the reviewer, where the measurements are concentrated closely enough that adjustments to the IQR range would not result in different estimations of the throughput. To show this, we have added an additional Figure 10 in Appendix F.2, which demonstrates that the estimated throughput would differ by a small amount even with a different IQR range.
>
> Additionally, in response to this feedback, we have included additional details in Appendix G regarding the exact choice of hyperparameters we choose along with some discussion on the chosen values and their robustness with respect to the performances. We thank the reviewer for their feedback.
>
> ---
>
> **[Continued Below]**

---

> ### Author Response · Authors · 2025-11-25
> **Response (Part 3/3)**
>
> > L237 assumes "All-Reduce operations involved in DP and TP, and point-to-point communications involved in PP. We consider a canonical ring/tree scaling with hierarchical aggregation such that intra- and interhost connections are separately modeled" -- do these assumptions matter for the ability to provide domain knowledge priors or do you expect OpPa to generalize to other settings?
>
> **The assumptions in L237 are not required for the validity of our domain knowledge prior belief**, but instead are only used as a simple model to capture a general trend regarding the communication costs in a parallel training process and analytically quantify a general prior belief. As discussed in Section 4.1, the prior belief is meant to be used in tandem with a Gaussian process which efficiently models the deviation between the prior belief and the true measurements. **We demonstrate this generalizability throughout the experiments**, which considers NN training on various hardware settings, both in single-host and multi-host setups with varying communication methods. In these settings, due to the exact communication protocols not being completely known, it is possible that the actual communication do not exactly match the assumptions used to construct our prior belief. Despite this, we find that our prior belief still contributes to good performances for OPPA across these different hardware settings, showing how applicable they are even if the prior belief may not be completely accurate.
>
> Additionally, **our GP-based surrogate function is still robust enough to model the throughput and memory usage even if the prior belief is not completely accurate**. Note that for the results in the main paper, **we are able to select good parallelism configuration already even though not all hyperparameters are explicitly considered by our prior belief**, showing the ability of OPPA to capture effects from unseen hyperparameters. Furthermore, we have **performed an experiment to tune the parallelism configuration on MoE models involving expert parallelization, which is also not explicitly modeled by our prior belief**. This set of experiments can be found in the newly-added Figure 19 in the updated version of the paper, showing that **despite the expert parallelism not being modeled in the prior belief, OPPA is still able to utilize the modeling ability of GP**s to learn the best PC compared to the other methods.
>
> We thank the reviewer for this comment, and will look to clarify these points further in the paper when space permits.
>
> ---
>
> > L294 "we assume the additional contribution is expected to have no additional bias on the estimates of the quantities" - Can you please elaborate on why this is justified.
>
> In the formulation for our surrogate model as presented in Equation 2, we assume that the prior belief provides the value for the general trend of the throughput. We assume that while the true throughput may deviate from the prior belief provided, given the prior belief is sufficiently accurate, the true throughput would still be "centered" around this prior belief.
>
> The additional contribution term aims at modeling this deviation from the prior belief, and so is assumed to be "centered" around zero to reflect our assumption on the true throughput. In our case, we choose to model this additional contribution by a zero-mean Gaussian process, whose values will be centered around zero and provide no additional "bias" for the final prediction, and letting the additional "bias" instead be captured by the prior belief. The combination of the prior belief with the additional contribution therefore gives rise to the overall Gaussian process as written in Equation 3.
>
> We appreciate the reviewer for their comment. In light of this, we have rephrased the line in the paper such that more clarity is provided for this assumption, as we have elaborated in this response.
>
> ---
>
> We again thank the reviewer for their review, and hope that our response is able to improve your evaluation of our paper.

---

> ### Comment · Reviewer_jA8L · 2025-11-26
>
> Thank you for your responses. They have clarified my understanding of this work and alleviated my main concern regarding the early termination approach. As a result, I am raising my score from 4 to 6.

---

### Official Review · Reviewer_Mo37 · 2025-10-31

**Soundness:** 3
**Presentation:** 2
**Contribution:** 2
**Rating:** 4
**Confidence:** 3

**Summary:**

The paper proposes OPPA, a Bayesian-optimization–based tuner for multi-dimensional parallel training configurations. OPPA combines a parallelism-informed prior, a GP residual to capture unmodeled effects, a constrained UCB to penalizes OOM, and a early-termination rule to stop clearly suboptimal trials. Experiments on BERT/Qwen/LLaMA across single- and multi-host clusters show higher achieved throughput over baselines.

**Strengths:**

- **Practical problem and clear formulation.** Treating parallelism configuration (PC) search as constrained black-box optimization with real hardware feedback is well motivated and useful for practitioners.
- **Principled surrogate design.** The additive “prior + GP residual” captures domain knowledge while remaining adaptable; the GP uncertainty naturally supports exploration.
- **Evidence across setups.** Results on 8/16/32-GPU settings (single/multi-host) and different transformer sizes show consistent throughput gains vs baselines; ablations isolate the value of the prior and early termination.

**Weaknesses:**

- **Incremental over prior autotuners.** Combining a hand-crafted prior with BO and early stopping is reasonable, but the conceptual advance over existing autotuners (e.g., FlexFlow/Alpa cost-model/simulator planners) appears modest. The paper should articulate why **BO is uniquely appropriate** for PC search—beyond empirical wins.
- **MoE/expert parallel not evaluated.** Given current practice, omitting expert-parallel (EP) MoE is a major gap. The prior does not model all-to-all traffic, routing imbalance, or token-dependent heteroscedasticity typical in MoE; early termination could be fragile under bursty load imbalance.
- **Early-stop sensitivity.** The theorem guarantees existence of ${βᵢ, τ_q}$, but practical tuning of $q_{min}, q_{max}, τ_q, βᵢ$ is nontrivial. Warm-up effects (graph capture, kernel autotuning, cache fills) can make early steps unrepresentative, risking premature termination and biased selection. The paper does not quantify false-negative stop rates.

**Questions:**

- The method introduces many hyperparameters. How are they tuned, and how sensitive are results to each? I recommend the authors to include a table listing all hyperparameters with definitions, default values, and search ranges.

---

> ### Author Response · Authors · 2025-11-25
> **Response (Part 1/2)**
>
> We thank the reviewer for their feedback, and for their positive feedback regarding the fact that we have proposed a "practical problem and clear formulation", along with the fact that OPPA utilizes a "principled surrogate design" and its benefits being proven with "evidence across setups".
>
> Below, we provide responses to  the points raised by the reviewer.
>
> ---
>
> > Combining a hand-crafted prior with BO and early stopping is reasonable, but the conceptual advance over existing autotuners (e.g., FlexFlow/Alpa cost-model/simulator planners) appears modest...
>
> Here, we refer to the discussion we have provided in the Related Works section of our paper, particularly the paragraph beginning on Line 147. As discussed there, we note that one of the key observations in our paper is that **the true throughput of a parallelism configuration cannot be specified exactly** since they are affected by various factors that may not be exactly known. This means that **existing cost models or simulations can only provide approximations of the true throughput**, and as a result **can deviate from the true throughput scores**.
>
> Since the true throughput cannot be specified exactly, the only method to know the true throughput of some parallelism configuration is by actually trialing them on real hardware. In this regard, **BO is uniquely appropriate for this problem since it is able to use the throughput measurements directly for the optimization process** without requiring the throughput to be modeled exactly, and adaptively select which configuration should be trialed next.
>
> However, naively using BO remains inefficient, as demonstrated in the experiments. To remedy this, OPPA **exploits the parallelism-informed prior belief** along with an **early termination mechanism to shorten suboptimal trials**, both of which allow for the efficiency of the BO algorithm to be significantly improved while maintaining sublinear regret in theory. In this regard, the empirical gains of OPPA over existing autotuners as presented in the experiments arise directly **from the effective incorporation of direct throughput measurements** along with **novel algorithmic advances which further boost its efficiency over existing methods**.
>
> ---
>
> > Given current practice, omitting expert-parallel (EP) MoE is a major gap. The prior does not model all-to-all traffic, routing imbalance, or token-dependent heteroscedasticity typical in MoE; early termination could be fragile under bursty load imbalance.
>
> We thank the reviewer for their feedback. In this regard, we have added an additional experiment in Figure 19 in Appendix I.3 of the updated paper which **attempts to tune the PC for MoE models**, which also includes the hyperparameters for the expert parallelism size. Note that **we did not change the prior belief function to incorporate expert parallelism size directly**.
>
> Here, we see that **OPPA is still able to outperform the other benchmarks and select a better PC compared to the other methods**. We **attribute this to our choice to use Gaussian process surrogate**, which is able to still model the unknown throughput well despite the prior belief not being completely accurate and even when different throughput estimations have different variances.
>
> We thank the reviewer for their suggestion regarding this experiment. As we have mentioned, we will incorporate these points into the updated version of our paper.
>
> ---
>
> **[Continued Below]**

---

> ### Author Response · Authors · 2025-11-25
> **Response (Part 2/2)**
>
> > The theorem guarantees existence of $\beta_i, \tau_q$ but practical tuning of $q_{min}, q_{max}, \tau_q, \beta_i$ is nontrivial.
>
> We note that in practice, fixing these hyperparameters to a certain value (as we recommend within the paper) is often sufficient to achieve good performances in all cases. This is partially due to the fact that certain ranges of the hyperparameter values are likely to make OPPA behave similarly anyway.
>
> For the value of $q_{max}$, this will determine the possible accuracy of the predicted throughput. As presented in the newly-added Figure 15, we see that after a certain point, larger values of $q_{max}$ has little effect on the measured throughput, meaning OPPA is quite robust to its choice. In practice, this could be set large enough such that the practitioner is satisfied with the throughput estimation as we have done for the benchmarks.
>
> For the value of $q_{min}$, we have provided ablation studies shown in Appendix I.7, where we see the effects of the chosen $q_{min}$ on the throughput obtained in the end. We notice a similar phenomenon that the reviewer mentions where the first few training steps are insufficient due to warm-up effects, which results in cases where $q_{min}$ is too small to result in poor performances. However, once $q_{min}$ is reasonably large, there is little effect in the throughput obtained at the end of the optimization. With this finding, we see that $q_{min}$ can be set to be quite small, but up to a certain point. In our experiments, we have set $q_{min} = 5$ for all experiments, which provides good results throughout all experiments.
>
> For $\beta_i$ and $\tau_q$, while we can alter the values to match the theoretical bounds, this is unnecessary in practice. In practice, we fix $\beta_i$ to one value (in our case, $\beta_i = 1$). Meanwhile, setting $\tau_q$ to a small enough number (in our case, $10^{-3}$) is sufficient to distinguish between suboptimal PCs. Both of these choices are as stated in Appendix F.3 in the original paper (and moved to Appendix F.3.3 in the updated paper). As the results in the paper suggest, this choice is already able to give strong performances compared to the other benchmarks.
>
> While we have already noted these information within the paper, we agree with the reviewer that they could be made more explicit, and will compile these recommendations as a summary in  Appendix G. We thank the reviewer for this feedback.
>
> ---
>
> > The method introduces many hyperparameters. How are they tuned, and how sensitive are results to each? I recommend the authors to include a table listing all hyperparameters with definitions, default values, and search ranges.
>
> We note that in Appendix C.1, **we have listed all the hyperparameters in a parallelism configuration** which we optimize over along with their feasible range, while in Appendix I.2 **we have also noted the chosen hyperparameters by OPPA**, along with some analysis of their sensitivity by comparing multiple trials run and based on the learned surrogate GP in Appendix I.4. We have included this information in the appendix due to space constraints, but will attempt to shift them into the main paper if space permits.
>
> ---
>
> We thank the reviewer for their feedback, and hope that our response is able to improve your evaluation of our paper.

---

> > ### Comment · Reviewer_Mo37 · 2025-11-26
> >
> > The reviewer thanks the authors for the additional experiments. My concerns regarding expert parallel and hyperparameter tuning have been alleviated. Now I tend to accept this work and encourage the authors to further extend OPPA under EP for broader impact.

---

### Official Review · Reviewer_r5so · 2025-11-03

**Soundness:** 4
**Presentation:** 3
**Contribution:** 2
**Rating:** 6
**Confidence:** 2

**Summary:**

This paper considers the design of parallel configurations for training neural nets with many GPUs.
It uses Bayesian optimization to maximize training throughput (training examples processed per unit time) subject to a constraint on GPU memory.   A UCB-style regret guarantee is given and experimental results are presented.

**Strengths:**

My background is in Bayesian optimization but not in AutoML / neural architecture search.

(Weakness) From a BO perspective, this paper is a straightforward application paper without a great deal of novelty.  A constrained BO problem is proposed --- maximize training throughput subject to a constraint on GPU memory use.  Informative priors are proposed that consist of carefully-crafted application-specific mean functions plus a mean-zero Gaussian process (this is sometimes called "universal kriging" in the old literature).  A UCB acquisition function is proposed and a regret guarantee is provided.  The regret guarantee appears to be fairly standard though the authors should inform me if there is something novel in the proof.  So from a BO perspective, the paper does not rise above the publication bar for ICLR.

(Strength) But from an AutoML perspective, the contribution seems valuable.  The method seems to outperform several benchmarks that seem reasonable to me by a practically meaningful margin in an evaluation that seems realistic.  I would be interested to hear from someone who has more experience with problems in this space who can comment on the selection of benchmark methods and the evaluation methodology.

**Weaknesses:**

See the text above under Strengths

**Questions:**

When evaluating parallel configurations, training instances are processed.  Are the results used or thrown away?

The paper is written as if we have some budget for choosing the parallel configuration, after which we get some reward. But it seems that this can be formulated as an online problem --- my goal is to process all of the training data as quickly as possible and I can switch parallel configurations during training.  Presumably there is some large overhead for switching parallel  configurations, so from a practical perspective any good algorithm will eventually pick one configuration and finish a large fraction of the training with this. Yet, it doesn't seem that the budget before you settle on one must be fixed. In this sense, the model in the paper seems a bit like it could be improved.

 Also, if the cost for switching configurations is large, then we this should be reflected as an additional cost during optimization of the parallel configuration.

---

> ### Author Response · Authors · 2025-11-25
> **Response (Part 1/2)**
>
> We thank the reviewer for their review, and their positive feedback regarding the valuable contribution of our work. Below, we provide responses to the points raised by the reviewer.
>
> ---
>
> > From a BO perspective, this paper is a straightforward application paper without a great deal of novelty...
>
> We would like to clarify that besides an effective introduction of a parallelism-informed prior belief, **another key contribution of our work (i.e., motivated by the nature of our parallelism configuration (PC) optimization problem) lies in the novel mechanism for early termination of suboptimal trials, which is integrated into our BO algorithm and exploited to significantly boost the efficiency of the BO algorithm while preserving its sublinear regret**. This early termination mechanism has been motivated in the Introduction and discussed in Section 4.3. We believe that the reviewer may have missed this contribution, as the **proposed early termination mechanism does not seem to be mentioned in the review**. Hence, we sincerely like to request the reviewer to take into account this contribution when deciding their final opinion and rating of this work.
>
> To elaborate, when optimizing the PC, to measure the throughput, a PC would be trialed by running a sequence of training steps and averaging the measurements across these repeated training steps. With these repeated steps, we consider whether fewer repeats could have been run while still recovering the optimal PC in the end. Motivated by this viewpoint, we propose an early termination mechanism which is proven to reduce the number of training steps required while also still recovering the optimum PC. We also note that this mechanism is useful, beyond PC optimization, in scenarios such as scientific experiments which may have to be conducted sequentially and repeated in order to obtain more accurate measurements of the true function value.
>
> Additionally, despite some trials being terminated earlier (and therefore having a less accurate measurement of the throughput), we are able to demonstrate that **the use of the early termination procedure is able to still guarantee sublinear cumulative regret in theory**, and that the optimal configuration can still be determined accurately since those cases will not be terminated early with high probability, both of which are a consequence of Theorem 4.1 and proven in Appendix F.3.2. The benefits of early termination is also empirically demonstrated via synthetic examples in Figure 14 and via real problem settings of optimizing parallel training throughout the results in the paper.
>
> We appreciate the reviewer for their comments. In response to this, we have adjusted the Introduction and Related Works sections to further highlight the novelty of OPPA from the BO standpoint as well.
>
> ---
>
> > ⁠...the selection of benchmark methods and the evaluation methodology.
>
> As mentioned in the second paragraph and throughout the Experiments section (Section 5), the benchmarks we chose to compare against OPPA are based on some of the methods that are used in practice. More specifically, we have compared OPPA against an adaptive approach which uses XGBoost as a surrogate model which is implemented in the AutoTuner module in DeepSpeed [1], which demonstrates inferior effiency compared to OPPA. Furthermore, we also compare OPPA with methods that use non-adaptive cost model surrogates, which are commonplace in the existing literature [2,3] as well, which is less effective than OPPA which uses the actual throughput measurements during the optimization process.
>
> In our results, we choose to report metrics that practitioners care about in practice and which correspond to physical measurements. Our experiments report the throughput when training on a chosen PC, which is a direct measure of how rapidly training can be done and is thus a good metric to reflect how good a PC is. We also directly report the actual time each method takes to find a good PC, which again is a very practical measure of performance that relates directly to a physical resource (i.e., actual time to run).
>
> ---
>
> [1] https://www.deepspeed.ai/tutorials/autotuning/
>
> [2] AMP: Automatically Finding Model Parallel Strategies with Heterogeneity Awareness. NeurIPS 2022.
>
> [3] nnScaler: Constraint-Guided Parallelization Plan Generation for Deep Learning Training. OSDI 2024.
>
> ---
>
> **[Continued Below]**

---

> ### Author Response · Authors · 2025-11-25
> **Response (Part 2/2)**
>
> > When evaluating parallel configurations, training instances are processed. Are the results used or thrown away?
>
> We note that **since the throughput is measured on real training steps, actual training instances can be processed in practice and used during training**. That said, the optimization process will usually process so few training steps compared to the full training anyway such that the **training steps performed during the optimization by OPPA is likely negligible compared to the full training process**. This is because the **number of training steps run per PC trial is small**, and would also be further reduced given that **OPPA will perform early termination for a large number of suboptimal training trials** anyway. Because of this, the training steps at the beginning could also be omitted in practice as well.
>
> More concretely, in Figure 5b, we have plotted the number of training steps that would be processed during the PC optimization and during the actual training. Note that here, OPPA only runs for 20 minutes while only trialing a few hundred training steps. This is negligible compared to the full training process that is ran for several hours, and would execute at least two orders of magnitude more training points.
>
> ---
>
>
> > The paper is written as if we have some budget for choosing the parallel configuration, after which we get some reward. But it seems that this can be formulated as an online problem --- my goal is to process all of the training data as quickly as possible and I can switch parallel configurations during training...
>
> We note that for the problem setting that the reviewer has mentioned where we want to process the training data as quickly as possible, **one good approach would be to find the best parallelism configuration (PC) as rapidly as possible, then continue to use the same PC for the remainder of the training process**. The reason for this is that sticking to a more suboptimal PC for an extended period of time would lead to fewer training steps being processed over the same period of time. So, to reduce this wastage, it is best to run on a good PC from the start even if extra costs on switching PC is incurred in the beginning. This is demonstrated in Figure 5b, where a small gain in throughput of PC selected by OPPA allows for an increase in the number of training steps that can be run during training.
>
> In this viewpoint, **OPPA would be an excellent method to achieve this goal, since it is adapted to find the best PC as quickly and efficiently as possible**. Though BO itself can find the best PC somewhat efficiently already,  with the additional early termination of suboptimal PCs and parallelism-informed prior belief, OPPA is able to boost this efficiency even further while still guaranteeing sublinear regret. This in practice allows OPPA to also find the optimal PC but extremely efficiently, being a novel method even for the problem setting the reviewer proposed.
>
> ---
>
> > Also, if the cost for switching configurations is large, then  this should be reflected as an additional cost during optimization of the parallel configuration.
>
> We note that in our experiments, we have already considered the time required to switch between different PCs. Specifically, throughout all experiments conducted (e.g., main results in Figure 4), we have **plotted the best throughput found as a function of the time the optimization process has been run for**, which would already include the time required to switch between different PCs. We see that when keeping in mind the additional overhead costs to trial a PC, OPPA is already more efficient than the other baselines.
>
> ---
>
> We thank the reviewer for their constructive review, and hope that our response is able to improve the opinion of the reviewer for our paper.

---

### Author Response · Authors · 2025-12-03
**Summary of Rebuttals (Part 1/2)**

We appreciate the efforts of the reviewers and the AC during this ICLR, especially under the current circumstances within this year's conference.

In particular, we would like to thank the reviewers for their constructive review, and for their **unanimous recommendation for acceptance of our paper either in their original review** (Reviewer r5so) **or after our provided clarifications** (Reviewers Mo37 and jA8L, who have stated their intentions to increase the paper's overall rating from 4 to 6). This is based on the general agreement that **our paper tackles a useful and important practical problem** of tuning configurations for parallel NN training, proposes a **novel and principled solution to solve said problem**, and provides **convincing empirical evidence of the method** in the experiments.

To provide further context for the discussion period thus far, we summarize some of the points raised by the reviewers and responses we have provided in the rebuttal period.

---

1. Comments regarding novelty of the paper. We note that **all reviewers have commented that OPPA tackles a useful and important practical problem** of speeding up parallel NN training. Despite this, some comments were raised by Reviewers r5so and jA8L regarding the methodological novelty of OPPA beyond the use of Bayesian optimization (BO) alone.
    - In our rebuttals, we have provided **clarifications on how OPPA leverages unique characteristics of the problem of optimization parallelism configurations** by **exploiting the parallelism-informed prior belief and early termination of suboptimal trials**, which result in **significant boost of OPPA's efficiency in practice** while still **preserving its sublinear regret**.
    - Additionally, to make our contributions clearer, we have edited our submission to **highlight the novel components we have integrated into OPPA in the Introduction and Related Works** (around Lines 85 and 192), particularly the early termination mechanism and parallelism-informed prior belief, and motivate why they are able to boost the performances of OPPA.

---

2. Comments regarding additional experiments. In addition to the NN training workloads already experimented on in the paper (for inputs in text and image domains), Reviewer Mo37 has advised us to run additional experiments for training mixture-of-experts (MoE) which can contain nuances within its computation that may not be fully captured by OPPA.
    - In response, we have added **Figure 19 in Appendix I.3** which **provides additional experiments for tuning the PC for mixture-of-experts model**, where expert parallelism is also optimized. Here, we see that OPPA is still able to generalize to unseen hyperparameters (in this case, expert parallelism dimension size) due to the flexible surrogate model within OPPA.
    - We note that these additional results have alleviated the concerns regarding additional experiments raised by Reviewer Mo37, who **has improved their evaluation of our work in response to the rebuttals**.

---

3. Comments regarding the early termination method. Reviewer jA8L has provided a comment suggesting that the early termination mechanism employed could be better justified in the paper.
    - As a response, we point out that within the original paper submission, we have provided **theoretical justification by showing that sublinear regret is preserved despite the early termination** (in Theorem 4.1 and Appendix F.3). Along with this, we also provide **empirical justification by showing that the exploitation of the early termination mechanism boosts efficiency of OPPA in practice** (Figures 4, 5, and 7).
    - In **Appendix F.3.1 and Figures 11-13** of the updated submission, we have also **provided additional intuition behind why the early termination is valid and helpful** especially in cases where repeated measurements are required (such as for optimizing parallelism configurations). This further justifies our design of the early termination mechanism via intuitive explanations.
    - The provided evidence and further intuition behind the early termination mechanism validates the improvement of OPPA based on its incorporation, which directly **strengthened the support of Reviewer jA8L as stated in their response to our rebuttals**.

---

**[Continued Below]**

---

> ### Author Response · Authors · 2025-12-03
> **Summary of Rebuttals (Part 2/2)**
>
> 4. Comments regarding the choice of hyperparameters. Reviewers Mo37 and r5so have raised questions on how the hyperparameters within OPPA are chosen in practice, and their effect on the overall algorithm.
>     - In our rebuttals, we noted that **the choice of these hyperparameter values has already been discussed in the original submission of the paper**. In particular, OPPA uses a fixed choice of $\beta_i$ and $\tau_q$ as stated in Appendix F.3. Meanwhile, the choice of $q_{min}$ has been studied by ablation studies in Appendix I.7, which shows that OPPA can converge to the optimal configuration as long as $q_{min}$ is sufficiently large.
>     - We have also **consolidated the choice of hyperparameter values used by OPPA in Appendix G**. Here, we list the hyperparameter values used by OPPA, and discuss how the choice of such values generally yields similar results within a certain range. The choice is backed by empirical evidence provided in the paper, which shows that OPPA performs similarly given that the hyperparameter values are within a certain reasonable range.
>
> ---
>
> 5. Comment regarding alternate settings of the problem. While the reviewers generally agree that our current problem setting is practical and realistic, Reviewer r5so has inquired about alternate settings for the problem where the optimization is done in an online setting such that the training data is processed simultaneously.
>     - In our rebuttals, we have highlighted that since OPPA already requires performing actual training steps anyway in order to obtain throughput measurements, it could directly be used in the setting proposed by the reviewer.
>     - Additionally, as we have clarified in our rebuttals, a potential method under the setting proposed by the reviewer would be to optimize the parallelism configuration as quickly as possible and then perform train on the remaining data with the optimal configuration. The early termination mechanism exploited by OPPA would also allow for futher speedup of the optimization process. In this case, we therefore argue that not only would OPPA be able to run in the setting proposed by the reviewer, but it would provide a competitive baseline in said setting as well.
>
> ---
>
> 6. Comments regarding cost model-based methods. Reviewer Mo37 believed that while OPPA shows empirical gains over existing cost model-based methods (as demonstrated in our experiments such as in Figure 8), these gains are not sufficiently articulated in the paper.
>     - In response, we have pointed out that **the reasons for these empirical gains have in fact been discussed in the Related Works section of our paper**, particularly the paragraph beginning on Line 147. Here, we explain that the true throughput of a parallelism configuration cannot be specified exactly since they are affected by various factors that may not be exactly known. This means that existing cost models or simulations can only provide approximations of the true throughput, and as a result can deviate from the true throughput scores.
>     - In this regard, Bayesian optimization is appropriate for this problem since it is able to use the throughput measurements directly for the optimization process. OPPA then further exploits the parallelism-informed prior belief along with an early termination mechanism to further boost the optimization process to obtain the performances reported in the paper.
>
> ---
>
> 7. Comments on assumption of the surrogate model. Reviewer jA8L has raised some minor issues regarding the assumptions of our surrogate model which is used by OPPA, particularly the assumed prior knowledge and the additional contribution terms.
>     - In response, we **clarify in the rebuttals that the prior knowledge is only aimed to capture a general trend regarding the communication costs in a parallel training process**, and that any deviation will not harm the overall model performance. To illustrate this, we have also pointed out that the experiments throughout the paper, including the MoE experiments that are added during the discussion period, considers very different hardware settings and optimizes hyperparameters not modeled in the prior belief. The strong empirical results provides evidence to justify our claim.
>     - Additionally, we have clarified in the rebuttals that we assume that while the true throughput may deviate from the prior belief provided, given the prior belief is sufficiently accurate, the true throughput would still be "centered" around this prior belief. The deviation is therefore moeled with a Gaussian process with zero mean which introduces "no additional bias". However, in light of the comment, we have added further clarification in the updated submission as hinted by the reviewer.
>
> ---
>
> We again thank the reviewers for their constructive review, and the AC for their hard work during this conference period.

---

### Meta-Review · Area_Chair_TVE5 · 2026-01-06

**Summary:**

This paper proposes OPPA, a Bayesian optimization approach for tuning parallel training configurations using real throughput measurements. The method combines a parallelism-informed prior, a GP residual model, explicit OOM constraints, and early termination to reduce wasted evaluations. Reviewers agree the problem is practically important and the experiments are solid, but the decision turns on whether the contribution is sufficiently novel beyond standard constrained BO plus careful systems-driven design.

**Reviewer Concerns:**

Concerns addressed by the rebuttal:
1) Coverage gaps: the rebuttal improves empirical coverage (including additional parallelism regimes) and answers several concrete questions about overhead and measurement protocol.
2) Clarity: the contribution and assumptions are presented more clearly after rebuttal, and the early-termination design is better justified.

Concerns still outstanding:
1) Novelty remains the main issue. Even after clarification, the method largely reads as an integration of known BO components (prior + GP residual + constraints) tailored to this application, rather than a distinctly new algorithmic idea.
2) Generality/transfer: it is still not fully established how easily the approach ports to different hardware stacks, frameworks, or rapidly changing system constraints without significant re-engineering of priors/constraints.
3) Practical comparison bar: while results are strong, it remains debatable whether the gains over existing autotuning practices justify acceptance at ICLR given the incremental methodological step.

**Reviewer Scores:**

1) Reviewer (score 6): likely unchanged at 6; their view is already “useful but not particularly novel,” and the rebuttal does not materially change that.
2) Reviewer (score 4): likely increase to 5; added coverage and clarifications reduce uncertainty but do not fully resolve novelty concerns.
3) Reviewer (score 4): likely increase to 6; rebuttal answers their concrete questions and improves presentation.

---

### Decision · Program_Chairs · 2026-01-26

Reject